

# Sedimentary response to sea ice and atmospheric variability over the instrumental period off Adélie Land, East Antarctica

P. Campagne[1,2,3,4], X. Crosta[1], S. Schmidt[1], M. N. Houssais[2], O. Ther[1], and G. Massé[2,3]

[1]EPOC, UMR CNRS 5805, Université de Bordeaux, Allée Geoffroy St Hilaire, 33615 Pessac, France
[2]LOCEAN, UMR CNRS/UPCM/IRD/MNHN 7159, Université Pierre et Marie Curie, 4 Place Jussieu, 75252 Paris, France
[3]TAKUVIK, UMI 3376 UL/CNRS, Département de Biologie, Université Laval, G1V 0A6 Quebec, Canada
[4]Québec-Océan, Université Laval, 1045 Avenue de la Médecine, G1V 0A6 Quebec, Canada

Received: 27 November 2015 – Accepted: 1 December 2015 – Published: 25 January 2015

Correspondence to: P. Campagne (p.campagne@epoc.u-bordeaux1.fr)

Published by Copernicus Publications on behalf of the European Geosciences Union.

**BGD**

doi:10.5194/bg-2015-610

Sedimentary response to sea ice and atmospheric variability

P. Campagne et al.

Title Page

| Abstract | Introduction |
| Conclusions | References |
| Tables | Figures |

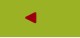 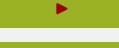

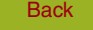 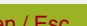



**BGD**

doi:10.5194/bg-2015-610

**Sedimentary response to sea ice and atmospheric variability**

P. Campagne et al.

**Abstract**

Diatoms account for a large proportion of primary productivity in Antarctic coastal and continental shelf zones. Diatoms, which have been used for a long time to infer past sea-surface conditions in the Southern Ocean, have recently been associated with di-
atom specific biomarkers (HBI). Our study is of the few sedimentary research projects on diatom ecology and associated biomarkers in the Antarctic seasonal sea ice zone. To date, the Adélie Land marginal ice zone has received little attention, despite evidence for the presence of high-resolution laminated sediment accumulation, allowing for finer climate reconstructions and sedimentary process studies. Here we pro-
vide a sequence of seasonally to annually laminated diatomaceous sediment from the DTCl2010 interface core retrieved on the continental shelf off Adélie Land, covering the 1970–2010 CE period. Investigations through statistical analyses of diatom communities, diatom specific biomarkers and major element abundances document the relationships between these proxies at an unprecedented resolution. Additionally, comparison
of sedimentary records to meteorological data monitored by automatic weather station and satellite derived sea ice concentrations help to refine the relationships between our proxies and environmental conditions over the last decades. Our results suggest a coupled interaction of the atmospheric and sea surface variability on sea ice seasonality, which acts as the proximal forcing of siliceous productivity at that scale.

## 1   Introduction

Diatoms have been used for a long time to infer past sea-surface conditions in the Southern Ocean on the basis of large-scale ecological studies from either plankton (e.g. Hasle, 1969) or surface sediments (DeFelice and Wise, 1981; Zielinski et al., 1997; Armand et al., 2005; Crosta et al., 2005). However, little is known about diatom
ecology in the seasonal sea ice zone and, especially, in the coastal and continental shelf zone (CCSZ) off East Antarctica. To date, the Adélie Land region has received

Interactive Discussion

Discussion Paper | Discussion Paper | Discussion Paper | Discussion Paper

**[BGD](BGD)**

doi:10.5194/bg-2015-610

**Sedimentary response to sea ice and atmospheric variability**

P. Campagne et al.

little attention, despite an abundance of evidence for the presence of high-resolution laminated sediment accumulation (e.g. Escutia et al., 2010). A very limited number of ecological studies have been performed at the species or species group level for plankton (Beans et al., 2008; Riaux-Gobin et al., 2013) or sediment (Leventer et al., 1992;
Maddison et al., 2006, 2012; Denis et al., 2006), and even fewer studies have investigated their relationships with local or regional environmental conditions. As a result, paleoenvironmental reconstructions for this area that are using diatoms are based on large scale studies and do not take into account the regional specificities. The Adélie Land region is of further particular interest due to the presence of coastal polynyas in
the vicinity of the Dumont d'Urville station (DDU) and the Mertz Glacier (MG), that are biologically very productive and where intense sea ice formation during winter leads to a large volume of dense water production (Arrigo and van Dijken, 2003; Sambrotto et al., 2003; Laccara et al., 2014).

Investigations of diatom communities, diatom specific biomarkers and major element
abundances at high resolution along a 72.5 cm long interface core retrieved in the Dumont d'Urville Trough (DDUT) allowed documentation of (1) the relationships between these proxies. The comparison of our proxy records to meteorological data inferred from either the automatic weather station (AWS) at DDU and satellite derived sea ice concentrations helped to determine (2) the relationships between environmental condi-
tions and proxy data, and consequently, the sedimentary response to atmospheric and sea surface changes. Here, we refine our knowledge of diatom ecology at the regional scale and propose a robust tool to infer past sea-surface conditions off Wilkes Land.

## 2 Environmental settings

### 2.1 Geographic features

The Dumont d'Urville Trough (Fig. 1), located along the Adélie Land on the East Antarctica margin is composed of several glacial depressions. These topographic features, up

to 1000 m deep, act as traps for sedimentary material (primary production and terrigenous particles) settling out from surface waters. The trough runs from the front of the Zélée and Astrolabe glaciers to the continental shelf break along a SE–NW orientation, and is bordered on its eastern side by the DDU Bank (Fig. 1), which culminates at 200 m below the sea level and limit exchanges with the Adélie Depression. On the western side, the DDUT is flanked by the Dibble Bank where the Dibble Ice Tongue develops northward.

## 2.2 Water masses

The Adélie Land is influenced by several water masses and currents (Rintoul, 1998; Williams and Bindoff, 2003; Williams et al., 2008). The wind-driven East Wind Drift (EWD) flows westward at the surface, and the Antarctic Surface water constitutes the summer sub-surface water mass on the continental shelf. The Circumpolar Deep Water (CDW) upwells near the Antarctic Divergence and intrudes onto the plateau during summer. The high-salinity shelf water, originates from the brine rejections during winter sea ice formation and from the cooling of the CDW (Fig. 1), and flows northward at the sea bottom as part of the dense shelf water that extrudes from the plateau. This Adélie Land Bottom Water ultimately sources the Antarctic Bottom Water (Rintoul et al., 1998; Jacobs et al., 2004; Meredith et al., 2013).

## 2.3 Wind conditions

The Adélie Land coast experiences the windiest conditions ever recorded on Earth through the presence of intense katabatic winds (Périard and Pettré, 1993) that are funneled by narrow glacial valleys close to the shoreline (Wendler et al., 1997). Although relatively strong winds ($> 10 \, \text{m s}^{-1}$) blow widely between 65 and 225° at DDU, the station is characterized by a dominant and recurrent katabatic wind from 140–180° (SE) coinciding with the maximum the wind speed ($> 25 \, \text{m s}^{-1}$) (Adolph and Wendler, 1995; Koning-Langlo et al., 1998). Such winds support the annual occurrence of polynyas in

Discussion Paper | Discussion Paper | Discussion Paper | Discussion Paper | Discussion Paper |

**BGD**

doi:10.5194/bg-2015-610

**Sedimentary response to sea ice and atmospheric variability**

P. Campagne et al.

the region, such as the DDU Polynya (DDUP; 66.11–139.31° E), and are important in sea ice production leading to dense water formation in winter (Adolphs and Wendler, 1995; Massom et al., 1998; Arrigo and van Dijken, 2003).

## 2.4 Sea ice conditions

In general, the sea ice melts every year between November/December and February/March, with sea ice concentrations greater than 80 % in winter and dropping below 20 % in summer (Arrigo and van Dijken, 2003). The Adélie Basin, and notably the core site (DDUT), experiences fast ice during the winter months that largely melts back to the coast during the summer season (Massom et al., 2003, 2009; Smith et al., 2011; Wang et al., 2014). At its maximum extent, the fast ice develops ∼ 100 km offshore between the Dibble Ice Tongue in the west to the Adélie Bank in the east (Massom et al., 2009; Smith et al., 2011). From the early spring to autumn, unstable fast ice conditions, characterized by several fast ice breakouts and re-formations as a result of lack of anchor points, generally occur in the DDUT (Massom et al., 2009; Smith et al., 2011), whereas the fast ice persists later in the season over the banks. A key factor of the formation, recurrence and persistence of this fast ice buttress, is the numerous grounded small icebergs on the Adélie and Dibble Banks that trap the passing pack ice and act as anchor points for fast ice formation (Massom et al., 2001; Giles et al., 2008; Smith et al., 2011), growing two ice promontories from either side of the core site. The presence of the fast ice buttress is therefore closely associated to the intense sea ice formation in the Mertz Glacier Polynya and westward advection within the EDW (Massom et al., 2009).

## 2.5 Atmospheric impacts on sea surface conditions

Sea ice cover in the area is subject to a strong interannual variability in terms of formation and retreat (Massom et al., 2009; Smith et al., 2011). Recent studies suggest the sea ice conditions are closely linked to fast-ice dynamics over the DDUT, which

**BGD**

doi:10.5194/bg-2015-610

**Sedimentary response to sea ice and atmospheric variability**

P. Campagne et al.

is largely depending to synoptic scale wind fields (Adolph and Wendler, 1995; Massom et al., 2003, 2009). Few studies have focused on the impacts of atmospheric and oceanic conditions on the regional sea ice. Most of these studies are either limited in time (Adolph and Wendler, 1995; Massom et al., 2009; Wang et al., 2014) or are too low in resolution to appropriately cover the Adélie Land region (Massom et al., 2013). In order to refine the relationships and response of sea-surface conditions to atmospheric forcing, Principal Component Analyses (PCA) were performed between seasonal meteorological parameters (Fig. 2; Sect. 3.4), recorded from AWS in the vicinity of DDU and satellite data over the core site (Fig. S1 in the Supplement). Studying the seasonal lagged response between meteorological parameters is beyond the scope of this study, and would necessitate higher frequency analyses. We therefore restricted our investigations to the statistical relationships between parameters at the intra-seasonal scale.

### 2.5.1 Case A: westerly winds dominated years

Overall, our observations suggest that years characterized by more westerly winds during spring-summer are associated with a delayed sea ice retreat over the DDUT (Table S1 in the Supplement; Note S1). Heavier sea ice conditions and late sea ice retreat in the area are related to the southward migration of the Antarctic Circumpolar Trough, which induces a prevalence of a westerly wind component (Heil et al., 2006; Parish and Bromwich, 2007; Massom et al., 2003, 2009; Wang et al., 2014). Westerly winds promote larger swells, advect pack ice into the region, and decrease the open water fraction (Massom et al., 2003; Heil et al., 2006; Smith et al., 2011; Wang et al., 2014; Zhai et al., 2015). However, while a more westerly wind regime retards the ice break-up, these warmer winds also delay sea ice formation and advance in autumn (Fig. 2; Table S1; Note S1). Heil et al. (2006) suggested that annual fast-ice duration will drastically reduce in the context of an increased cyclonicity. This phenomenon is particularly evident in autumn, as the young thin ice is easily removed during storms, delaying the sea ice advance and fast ice formation. Repeated removal of fast ice under

**BGD**

doi:10.5194/bg-2015-610

**Sedimentary response to sea ice and atmospheric variability**

P. Campagne et al.

strong cyclonic warm winds would further lead to thinning the ice cover and an overall increase in sea ice production and northward export (Heil et al., 2006).

### 2.5.2 Case B: easterly winds dominated years

In contrast, during years that are characterized by stronger easterly winds during the spring-summer period, we observe a sea ice retreat earlier in the season (Table S1; Note S1). Massom et al. (2009) have suggested that easterly winds promote the presence of pack ice in the region but also contribute to building a fast ice buttress over the banks. Persistence of fast ice promontories during the summer over the Dibble and the DDU banks, can in turn act as a barrier to westward sea ice advection and lead to the establishment of open water conditions over the DDUT, as observed in several locations off East Antarctica (e.g. MGP and Terra Nova Bay polynyas) (Arrigo and van Dijken, 2003; Massom et al., 2001, 2009; Campagne et al., 2015). In autumn, we observe colder easterly winds, which promote an early sea ice advance. In contrast to westerly winds (Heil et al., 2006), easterly winds tend to increase ice advection and icebergs in the area as observed for the spring-summer period, providing favourable conditions for the formation of thick fast ice and buttress earlier in the autumn.

### 2.5.3 Case C: the role of katabatics

Although our PCA suggest that sea ice concentration increase with an increasing wind direction, from southerly-dominated to westerly-dominated wind fields (see above), the relationship between atmospheric and sea surface conditions is not linear. Indeed, strong and persistent southerly (katabatic) winds break the fast ice and promote the formation of a coastal polynya despite a sizably reduced buttress (Massom et al., 2009; Riaux Gobin et al., 2013). Such polynya regime is however constrained to the coastline (Vaillancourt et al., 2003), implying that open conditions may not extent over the core site, where the pack ice would be present due to a reduced fast ice buttress.

Discussion Paper | Discussion Paper | Discussion Paper | Discussion Paper |

**BGD**

doi:10.5194/bg-2015-610

**Sedimentary response to sea ice and atmospheric variability**

P. Campagne et al.

# 3 Material and methods

## 3.1 Core description and $^{210}$Pb chronology

A 72.5 cm long interface core, DTCI2010, was retrieved aboard the R/V *Astrolabe* (66°24.68′ S; 140°26.67′ E; 1010 m water depth) during the 2010 ALBION-HOLOCLIP

cruise. Positive X-ray images performed on the SCOPIX image-processing tool (Migeon et al., 1999) gave detailed information about sediment density and structure. SCOPIX images revealed laminations along the entire sedimentary section. The core was sampled continuously at 0.5 cm resolution and its chronological framework was determined based on $^{210}$Pb excess ($^{210}$Pb$_{xs}$; $T_{1/2}$ = 22.3 years). The activities of $^{210}$Pb

and $^{226}$Ra were measured on dried sediments by non-destructive gamma spectrometry using a well-type, high efficiency low background detector equipped with a Cryo-cycle (CANBERRA) (Schmidt and De Deckker, 2015). Activities are expressed in mBq g$^{-1}$ and errors are based on 1 standard-deviation counting statistics. $^{210}$Pb$_{xs}$ was determined by subtracting the activity supported by its parent isotope, $^{226}$Ra, from the total

$^{210}$Pb activity in the sediment. In the interface core DTCI2010, there is a general downcore trend in decreasing $^{210}$Pb$_{xs}$ activities, from 242 to 74 mBq g$^{-1}$, as expected due to the decay of the unsupported $^{210}$Pb (Fig. 3). However, the observation of a layer where activities decrease slowly between 10 and 40 cm led us to retain the Constant Initial Concentration model (CIC; Robbins and Eglington, 1975) to calculate the age

(t) of each measured horizon as: $t = (1/\lambda) \ln (A0 \, Az^{-1})$ where $t$ and Az are the age of the sediment and the excess $^{210}$Pb activity at the depth $z$, A0 is the activity at the surface and $\lambda$ is the decay constant of $^{210}$Pb. This choice is supported by the almost constant activities measured in the uppermost sediments of cores collected in 2003 (240 ± 13 mBq g$^{-1}$; Massé et al., 2011) and 2011 (253 ± 13 mBq g$^{-1}$ in DTCI2011;

Schmidt S., unpublished data). A second-order polynomial function was calculated from the 11 $^{210}$Pb-dates obtained over the entire core to build the age model (Fig. 3).

Discussion Paper | Discussion Paper | Discussion Paper | Discussion Paper | Discussion Paper |

**BGD**

doi:10.5194/bg-2015-610

**Sedimentary response to sea ice and atmospheric variability**

P. Campagne et al.



## 3.2 Diatom analyses

Micropaleontological analyses were performed according to the methodology described in Crosta and Koç (2007) at a ∼ 0.25 year resolution (every 0.5 cm) in core DTCI2010. Diatom slides were prepared from ∼ 0.5 g of dry sediment. A few drops of the resulting residue suspended in distilled water were evaporated onto a coverslip, which was subsequently mounted on a glass slide with NOA61. Counts were performed under a microscope at a magnification of ×1000. For each sample, 300–350 diatom valves were counted and data are presented as species relative abundances. Diatom identification was performed to the species or species group level. More details about slide preparation and diatom identification are available in Crosta et al. (2004). Nearly 70 diatom species were identified in down-core assemblages, from which 25 presented abundances higher than 2 % of the total diatom assemblages.

## 3.3 Biomarker analyses

Biomarker analysis followed the technique described by Massé et al. (2011) and were also performed at a ∼ 0.25 year resolution (every 0.5 cm) in core DTCI 2010. Briefly, an internal standard was added to 1 g of the freeze-dried sediments, lipids were extracted using a Dichloromethane/Methanol mixture to yield a total organic extract (TOE), which was then purified using open column chromatography (silica). Hydrocarbons were analyzed using a Gas Chromatograph coupled to a Mass Spectrometry detector (GC-MS). More details about analytical parameters are available in Massé et al. (2011).

## 3.4 Instrumental data

Daily sea ice concentrations (SIC) for the time period 1978–2012 were obtained from the National Snow and Ice Data Center data repository. The dataset is based on passive microwave observations from the Nimbus-7 SSMR (1978–1987) and DMSP SSM/I (1987–2007) and SSMIS (2007–2012) radiometers processed with the NASA Team

Discussion Paper | Discussion Paper | Discussion Paper | Discussion Paper

**BGD**

doi:10.5194/bg-2015-610

**Sedimentary response to sea ice and atmospheric variability**

P. Campagne et al.

algorithm (Cavalieri et al., 1995) at a spatial resolution of 25 km × 25 km. Averaged sea ice concentrations (SIC) were calculated over the core site in the central part of the DDUT (Fig. S1). Daily measurements of wind direction (from 0 to 360°), velocity (m s⁻¹) and temperature (°C) for the time period 1956–2011 were obtained from the METEO France publithèque. The dataset is based on AWS at the DDU French station (66.7° S, 140° E), established on Petrel Island approximately 2 km offshore (Wendler et al., 1997) and 30 km from the core site. The dataset provides monthly statistics based on 6 hourly in-situ observations at 10 m. Wendler et al. (1997) showed that from Penguin Point (east of the Mertz Glacier) to DDU stations, meteorological parameters displayed similar variations, leading to the conclusion that measurements from AWS were relatively robust. For both, satellite and AWS datasets, monthly anomalies are expressed relative to the mean value calculated for each month over the 1979–2009 period. PCA between meteorological parameters were performed at seasonal scale, spring (September to November), summer (December to February), autumn (March to May) and winter (June to August), while PCA between sedimentary data and meteorological parameters were averaged over the November to March period.

The sea ice retreat date was determined as the Julian day when the SIC (7 day average) dropped below 40 %, while the sea ice advance corresponds to the day when SIC increased to above 40 %. The duration of the ice-free season corresponds to the number of days per year during which SIC < 40 %. Wind direction represents the origin of the wind over 360°. East/South/North/Westerly wind components indicate the number of days between November and March during which the wind blows between 45–135°/135–225°/315–45°/225–315° respectively. A ratio between easterly and southerly wind components (hereafter noted E/S), and a ratio between northerly and westerly wind components (hereafter noted N/W), were also calculated and used to account respectively of the dominant and the secondary wind directions in the region.

Discussion Paper | Discussion Paper | Discussion Paper | Discussion Paper |

**BGD**

doi:10.5194/bg-2015-610

**Sedimentary response to sea ice and atmospheric variability**

P. Campagne et al.

## 3.5 Principal component analyses

Statistical analyses were run using the statistical software XLStat (Addinsoft). PCA and Pearson correlation tests (significance level $\alpha = 0.05$) were performed on 67 diatom species, 2 diatom specific biomarkers along with their associated ratio and on major elements to first (1) determine the relationships between species and identify potential diatom cluster (in order to increase the statistical weight of minor species) and (2) investigate the relationships between the different proxies in our study. The statistical analyses performed on all diatom species are presented in Sect. 4.1 and in Note S2, which includes the species accounting for less than 1 % of the total diatom assemblage. In a second PCA, significant sedimentary data, in terms of population and ecological preferences, were interpolated at one year and compared with meteorological data to determine how phytoplankton was associated with selected environmental parameters. In parallel, PCA and the Pearson correlation test were performed on seasonally averaged meteorological and satellite data, along with seasonally averaged values of SOI and SAM index, to support previous oceanographic and atmospheric observations about climate forcing and their environmental response in the area.

## 4 Results and discussion

Several diatom species in polar and sub-polar marine environments exhibit a narrow range of environmental preferences, especially in terms of sea ice conditions. Diatom assemblages are therefore commonly used in association with geochemical proxies to infer past sea-surface conditions. The distributions of these proxies have been studied and validated at synoptic scale throughout the Southern Ocean (e.g. Gersonde and Zielinksi, 2000; Crosta et al., 2005; Armand et al., 2005) but less is known about the Antarctic Coastal and Continental Shelf Zone (CCSZ). Off the Adélie-Georges V Land, it has been suggested that diatom communities primarily respond to water column stability and sea ice conditions and dynamics which in turn respond to atmospheric and

Discussion Paper | Discussion Paper | Discussion Paper | Discussion Paper |

**BGD**

doi:10.5194/bg-2015-610

**Sedimentary response to sea ice and atmospheric variability**

P. Campagne et al.

oceanic forcing (e.g. Beans et al., 2008; Riaux Gobin et al., 2005, 2013). In Sect. 4.1, we refine the use of diatoms and geochemical proxies off Adélie Land by statistically comparing their downcore abundances in DTCI2010. In Sect. 4.2, we study their relationships and response to regionally monitored environmental parameters. In Sect. 4.3, we discuss the relevance of our proxies as potential environmental indicators for paleoclimate reconstructions, by interpreting the evolution of our results over 40 years of instrumental records.

## 4.1 Relationship between sedimentary proxies

The first PCA (Fig. 4) includes all diatom species and geochemical proxies in order to observe their relationships and identify diatom clusters as groups of species having similar records in core DTCI2010. Cluster composition is based on significant coefficient correlation between the diatom species (see Sect. 3.5) combined to their known ecological preferences in the literature. Clusters allow the use of low abundance species ($< 2\%$; represented in grey on the first PCA; Fig. 4) that may bear important ecological signatures but cannot be used alone as environmental indicators. Only significant Pearson correlation coefficients are detailed in the following section. High level but non-significant Pearson coefficients are mentioned as strong relationships. Hereafter, we focus on the main proxies, in terms of abundances and ecological preferences. Results are summarized and presented in Table 1, and information on secondary proxies can be found in Note S2.

### 4.1.1 Sea ice related proxies

Antarctic sea ice diatom assemblages are largely represented by *Fragilariopsis* species with *F. curta* and *F. cylindrus* being the dominant taxa. Sediment trap studies indicate that relative abundances of the two species in the phytoplankton increase southward with increasing sea ice cover and decreasing temperature (Gersonde and Zielinski, 2000) with highest occurrences in stratified, ice melt influenced waters (Kang and Fryx-

Discussion Paper | Discussion Paper | Discussion Paper | Discussion Paper

**BGD**

doi:10.5194/bg-2015-610

**Sedimentary response to sea ice and atmospheric variability**

P. Campagne et al.

Discussion Paper | Discussion Paper | Discussion Paper | Discussion Paper |

**BGD**

doi:10.5194/bg-2015-610

**Sedimentary response to sea ice and atmospheric variability**

P. Campagne et al.

ell, 1992). This distribution is reflected in surface sediments. Highest abundances of *F. curta* occur in locations that experience 9–11 months yr$^{-1}$ of sea ice cover, with highly consolidated ice conditions (65–90 %) during winter, while *F. cylindrus* optimum is at 8.5 months yr$^{-1}$ and 70–90 % of winter sea ice concentration (Armand et al., 2005).

In the study area, investigation of the sediment microstructure demonstrates highest abundances of *F. curta* and *F. cylindrus* at the beginning of the annual couplet indicating an early growth phase and subsequent deposition (Maddison et al., 2006; Denis et al., 2006). As such, *F. curta* and *F. cylindrus* along with *F. vanheurckii* were often considered as indicators of heavy sea ice cover in several paleoclimate studies (Barcena

et al., 2002; Barbara et al., 2010), notably in our region (Crosta et al., 2007; Denis et al., 2010). *Fragilariopsis cylindrus* and *F. curta* are among the most abundant species in DTCI2010, while *F. vanheurckii* is not significant in our dataset and is not discussed hereafter (Fig. 4; Table S2 in the Supplement). *Fragilariopsis cylindrus* is located in the F1-&F2- quarter (Fig. 4), and displays a significant negative correlation with *F. kergue-*

*lensis* (−0.432; Table S2) and diatoms of clusters 4 and 6 (e.g. *T. lentiginosa*, −0.378; Table S2). However, *F. cylindrus* is also negatively and significantly correlated with *F. curta* (−0.306; Table S2). *Fragilariopsis curta* is located on F1+&F2- axes and is not significant (< 0.25) neither on F1 nor F2, suggesting this species is poorly sensitive to changes in the main environmental parameters at that timescale (Fig. 4). Large

changes in environmental conditions are probably necessary to induce a sedimentary response of *F. curta*.

Few marine and freshwater diatoms belonging to *Haslea*, *Navicula*, *Pleurosigma* and *Rhizosolenia* genera were recently found to be synthesizing Highly Branched Isoprenoids (HBI) (Sinninghé et al., 2004; Massé et al., 2011). A di-unsaturated isomer

[HBI:2] has been identified in Antarctic sea ice and isotopic analyses provide evidence for that this isomer is synthesized by sea ice dwelling diatoms, while a tri unsaturated isomer [HBI:3] is synthesized by phytoplankton diatoms (Collins et al., 2013). In Adélie Land, relatively high concentrations of [HBI:2] have been found to occur during the spring sea ice melt (Massé et al., 2011). Recent studies have proposed the use of

**BGD**

doi:10.5194/bg-2015-610

**Sedimentary response to sea ice and atmospheric variability**

P. Campagne et al.

[HBI:2] and/or [HBI:2]/[HBI:3] to reconstruct variations of Holocene Antarctic sea ice duration as a complementary approach to diatom counts (Denis et al., 2010; Collins et al., 2013). More regionally, Campagne et al. (2015) built on the co occurrence of [HBI:2] and *F. cylindrus* to infer periods of heavier sea ice conditions over the last 250 years in Commonwealth Bay. [HBI:2] and [HBI:2]/[HBI:3] are both located in the F1+&F2+ quarter, opposite [HBI:3] (Fig. 4), which agrees with the literature. However, [HBI:2] is significantly correlated with Ti (0.372; Table S2) and *T. antarctica* (0.218; Table S2), and does not show any relationships with the *Banquisia* gp. or *F. cylindrus,* unlike observations at longer timescales (e.g. Denis et al., 2010; Etourneau et al., 2013; Collins et al., 2013; Campagne et al., 2015). These results may suggest that [HBI:2] respond to different sea ice conditions than sea-ice related diatoms at the seasonal-annual scale, possibly because of different origin of production. However, differential export towards the sediment may also account for the observed differences (Collins et al., 2013).

*Thalassiosira antarctica* is mainly present in stratified Antarctic inshore waters (Johansen and Fryxell, 1985). In the Weddell Sea, *T. antarctica* blooms are observed in newly formed platelet ice in polynyas and in crack pools formed by disintegrating sea ice during summer (Gleitz et al., 1996). *Thalassiosira antarctica* is closely related to sea ice formation and/or breakup, as it blooms in open waters during summer-autumn, and produces resting spores (RS) at the end of the growing season when sea ice returns (Cunningham and Leventer, 1998, 1999). Taylor (1999) suggests that the formation of *T. antarctica* spores could be triggered by the low light intensities that occur beneath developing pack and platelet ice. Reduced wind mixing below the sea ice may also induce spore formation (Taylor et al., 2001). *Thalassiosira antarctica* RS, the main form encountered in sediments, is most abundant in regions where sea ice is present for at least 6 months $yr^{-1}$, and is believed to be induced under nutrient-stressed conditions or low light intensities (Armand et al., 2005). To note that the lower threshold of 6 months $yr^{-1}$ is attributable to the warm variety thriving in the northern Antarctic Peninsula (Taylor and McMinn, 2001), whereas the cold variety occurs mainly in

southern Antarctic Peninsula and coastal Antarctic zones where sea ice duration is above 8 months yr$^{-1}$ (Denis et al., 2006; Maddison et al., 2012). In DTCI2010, most valves were thus *T. antarctica* RS variety T1. In the Holocene sediment in the region, *T. antarctica* RS were found to co-occur with several large centric diatom species and

5 *F. kerguelensis* (e.g. Denis et al., 2006). *Thalassiosira antarctica* were found to share globally similar sea surface temperature, sea surface salinity, sea ice proximity preferences and similar seasonal occurrences with *Porosira glacialis* (Pike et al., 2009). On the PCA, *T. antarctica* (here as RS of the cold form) is located in the F1+&F2+ axes (Fig. 4). The species shows significant positive correlation with large centric diatoms

(e.g. *Thalassiosira lentiginosa*: 0.266; *Thalassiosira oliverana*: 0.280; Table S2) and higher correlation with *P. glacialis* (0.356; Table S2), in line with previous studies.

The *Porosira* group is composed of *P. glacialis* and *P. pseudodenticulata*. *Porosira glacialis* is associated to cold coastal waters adjacent to sea ice (Taylor et al., 1997; Zielinski and Gersonde, 1997). It has been observed in waters with high concentrations

of slush and wave-exposed shore ice (Krebs et al., 1987). *Porosira pseudodenticulata* is commonly observed in pack ice and fast ice. *Porosira* spp. are known to survive environmental stress (nutrient depletion, prolonged periods of darkness) by forming resting spores (RS) at the end of the ice-free season (Taylor and McMinn, 2001; Cremer et al., 2003; Pike et al., 2009). These spores can be incorporated into waxing sea

ice. As such, *Porosira* spp. can be directly seeded from the sea ice during spring of the following year (Gleitz et al., 1996). *Porosira glacialis* is abundant in regions that experience at least 7.5 months yr$^{-1}$ of sea ice cover (slightly longer than *T. antarctica*), with 30 % of summer sea ice concentration and highly compacted winter sea ice (65–85 % concentration) (Armand et al., 2005). As for *T. antarctica*, *P. glacialis* RS

sublaminae have been interpreted in Holocene laminated sediments in Adélie Land, indicating a late summer/autumn rapid deposition that is linked to early sea ice return (Maddison et al., 2006). *Porosira glacialis* and *P. pseudodenticulata* are respectively situated on the F1+&F2+ and F1+&F2- axes (Fig. 4). Pearson coefficient correlation

Discussion Paper | Discussion Paper | Discussion Paper | Discussion Paper | Discussion Paper |

**BGD**

doi:10.5194/bg-2015-610

**Sedimentary response to sea ice and atmospheric variability**

P. Campagne et al.

between these two species is significant (0.252; Table S2), arguing that both species can be grouped together in our study and confirming previous observations.

Other sea ice related proxies, the Banquisia gp. (*Navicula directa*, *N. glaciei*, *Synedra* spp. and *Ephemera* spp.), *Fragilariopsis obliquecostata*, *Eucampia antarctica*, and the *Fragilariopsis* summer gp. (*Fragilariopsis ritscheri* and *F. sublinearis*) are sometime used in the literature to infer past sea ice conditions but are not significant in our study and therefore are only discussed in Note S2.

### 4.1.2 Open ocean proxies

*Fragilariopsis kerguelensis* dominates phytoplankton assemblages of the open ocean zone south of the Polar Front where sea ice is absent during summer (Halse et al., 1969; Froneman et al., 1995) and thus characterizes open water conditions during summer. *Fragilariopsis kerguelensis* is observed in recent sediments of areas experiencing up to $8\,\text{months}\,\text{yr}^{-1}$ of sea-ice cover (Crosta et al., 2005). In several paleoclimate studies (e.g. Crosta et al., 2008), notably from the area, *F. kerguelensis*, present an inverse relationship to sea ice and cold-water species. Summer-autumn laminae were found to present high occurrences of *F. kerguelensis*, along with *T. antarctica* and large centric species (Denis et al., 2006; Maddison et al., 2006). On the PCA, *F. kerguelensis* is located on the F1+&F2- (Fig. 4), and displays a significant positive correlation with several large centric diatoms (e.g. *T. lentiginosa*, 0.507; *T. antarctica*: 0.184; *P. glacialis*: 0.235; Table S2), and a strong negative correlation with *F. cylindrus* ($-0.432$; Table S2) and *Chaetoceros* RS ($-0.248$; Table S2), in agreement with previous studies.

The Open Water group is composed of large centric diatoms. In general, species belonging to the genus *Thalassiosira* are most commonly found in areas experiencing open water conditions during the growing season (Johanssen and Fryxell, 1985). *Thalassiosira lentiginosa* and *T. oliverana* commonly occur in the Southern Ocean, south of the Polar Front, in areas characterized by permanent open ocean conditions (Johanssen and Fryxell, 1985). Relative abundances in sediment of *T. lentiginosa* show an inverse relationship with sea ice cover with high occurrences in areas experiencing be-

Discussion Paper | Discussion Paper | Discussion Paper | Discussion Paper |

**BGD**

doi:10.5194/bg-2015-610

**Sedimentary response to sea ice and atmospheric variability**

P. Campagne et al.

tween 0 and 4 months yr$^{-1}$ of sea ice presence and a decline towards prolonged sea ice duration (Crosta et al., 2005). *Thalassiosira oliverana* is clearly dominant in locations where open ocean conditions occur close to the sea ice edge during summer (Crosta et al., 2005). For *Thalassiosira gracilis*, the distinction between the two varieties (*T. gra-*
*cilis* var. *gracilis* and *T. gracilis* var. *expecta*) has not been performed in this study as this species presents low abundances in Antarctic coastal areas (Armand et al., 2005). In summer, the species appears most highly associated with conditions related to open-ocean conditions and its population diminishes in regions of unconsolidated sea ice (< 40 % concentration, SST below 2 °C) (Crosta et al., 2005). *Actinocyclus actinochilus*
has been observed with abundances over 2 % (which is not the case in our study area) in regions where sea ice cover persists more than 7 months yr$^{-1}$ (Armand et al., 2005). *Actinocyclus actinochilus* can be considered as a cold-water Antarctic species, and increasing sedimentary abundances of this species are in line with an ice-free region during summer (< 40 % concentration) and a strongly compact sea ice covered region in
winter (70–90 % concentration) (Armand et al., 2005). *Coscinodiscus* is generally considered as an open water genus (Garrison and Buck, 1989; Moisan and Fryxell, 1993). High abundances of *Coscinodiscus* spp. over the shelf were related to a southward influx of warm surface water (Taylor and Sjunneskog, 2002). *Stellarima microtrias* is reported as restricted to the Antarctic Zone south of the Polar Front (Zielinski and Ger-
sonde, 1997). The species has been observed attached to sea ice in waters influenced by the ice (Hasle, 1988). *Stellarima microtrias* is rarely used in paleoenvironmental studies as its ecology is poorly documented, its ecological affinity is uncertain and its abundances are generally low (Taylor et al., 2001). *Thalassiosira tumida* has maximum occurrences south of the Polar Frontal Zone, and is most abundant during the austral
summer (Cunningham and Leventer, 1998; Armand et al., 2005). Armand et al. (2005) indicate that maximum abundances of *T. tumida* occur in regions with low sea-ice cover during summer (open water conditions). Finally, Armand et al. (1997) have suggested that *Thalassiosira trifulta* thrives in the coldest waters amongst the *Thalassiosira spp.*. However, its overall low abundance precluded any use downcore alone. *Thalassiosira*

Discussion Paper | Discussion Paper | Discussion Paper | Discussion Paper | Discussion Paper |

**BGD**

doi:10.5194/bg-2015-610

**Sedimentary response to sea ice and atmospheric variability**

P. Campagne et al.

Interactive Discussion

**[BGD]**

doi:10.5194/bg-2015-610

**Sedimentary response to sea ice and atmospheric variability**

P. Campagne et al.

*lentiginosa*, *T. oliverana*, *A. actinochilus* and *S. microtrias* are localized on F1+&F2+, and *T. trifulta*, *T. gracilis*, *Coscinodiscus* spp., *T. gracilis* and *T. tumida* are localized on F1+&F2- (Fig. 4). *Thalassiosira lentiginosa* exhibits significant positive correlation with *Coscinodiscus* spp. (0.190; Table S2), *T. oliverana* (0.251; Table S2), *S.microtrias*
(0.186; Table S2), *A. actinochilus* (0.330; Table S2) and *T. gracilis* (0.266; Table S2). *Thalassiosira oliverana* also displays significant positive correlation with *A. actinochilus* (0.166; Table S2), *T.trifulta* is significantly correlated with *Coscinodiscus* spp. (0.174; Table S2) and *T. gracilis* is correlated with *A. actinochilus* (0.218; Table S2). *Coscinodiscus* spp. exhibits a significant coefficient correlation with *A. actinochilus* (0.244;
Table S2) and *S. microtrias* (0.211; Table S2). Strangely, *T. tumida* does not display a significant correlation with the species mentioned above despite being close to them in the PCA (Fig. 4). This species was therefore not included in the Open water group despite its distribution on the PCA and its documented ecological preferences. Our results show a significant positive correlation between *T. gracilis* (0.485; Table S2) and
*T. lentiginosa* (0.507; Table S2) with *F. kerguelensis*, confirming grouping in previous studies (e.g. Crosta et al., 2008). As with other clusters cited above (e.g. clusters 4 and 5), it seems that the Open Water gp. is also closely linked to the F1+ area in our analyses. We thereby decided to not include *Asteromphalus hookeri* in the group despite the few correlations (Table S2), as it seems to rather depend on F2 (Fig. 4). Ad-
ditionally, *Thalassiosira ritscheri*, *Actinocyclus curvatulus* and *Asteromphalus hyalinus* are insignificant in the F1+ area (Fig. 4) and, since they do not present any significant positive correlation with the other species (Table S2), we decided to remove them from the Open Water gp. *Thalassiosira gavida* also does not appear in this cluster as it is uncorrelated and positioned on the F1- area (Fig. 4). *Eucampia antarctica,* which is
significantly correlated to species of the Open Water gp., was separated given its high abundance.

[HBI:3] is positioned in the F1-&F2- axes, opposite to [HBI:2] and [HBI:2]/[HBI:3] (Fig. 4). This suggests that the [HBI:2] and [HBI:3] are produced in contrasting environments, in agreement with previous studies that proposed the use of [HBI:2] and [HBI:3]

to reconstruct variations of past sea ice cover as complementary sea ice proxy to diatom counts (Denis et al., 2010; Collins et al., 2013). Strangely, the Pearson correlation between [HBI:3] and [HBI:2] does not show any relationships in our PCA, (Table S2). This could be caused by the variability of the sea ice seasonality in our study area whereby Collins et al. (2013) suggested that [HBI:3] and [HBI:2] showed that these biomarkers correlate most positively and, consequently, co-occurred during a period of low seasonal sea ice change, and were less positively correlated during a period characterized by high seasonal change. Therefore, it seems that both biomarkers have to be considered carefully when they are taken separately. PCA show [HBI:3] is significantly correlated with *Rhizosolenia* spp. and *Proboscia inermis* (0.491 and 0.635 respectively; Table S2), in agreement with Sinninghé et al. (2004) that found the [HBI:3] to be synthesized by the *Rhizosolenoids*. While the analysis of the hydrocarbon content of several temperate species belonging to the genus *Proboscia* did not reveal the presence of HBIs, the polar species *P. inermis* may be able to synthesize these compounds.

Titanium (Ti) is considered as an indicator of terrigenous inputs (Denis et al., 2006; Presti et al., 2011). In Antarctic coastal areas, delivery of terrigenous particles is possible via several dominant modes such as meltwater discharge, ice rafting, runoff from outlet glaciers, and although, considered negligible in coastal East Antarctic regions, eolian transport (Presti et al., 2003; Escutia et al., 2003; Denis et al., 2006). In the literature, Ti content is used to infer past changes in terrigenous supply to the ocean and associated to open conditions (Denis et al., 2006; Campagne et al., 2015). Therefore, Ti is an indicator of enhanced melting period over the ice-free season. In our data, Ti is located in the F1+&F2+ area (Fig. 4), and displays high significant positive correlation with some open water taxa, e.g. *F. kerguelensis* (0.248; Table S2), *P. glacialis* (0.228; Table S2) and *T. antarctica* (0.238; Table S2), in line with previous studies. In the same way, Ti presents significant negative correlations with *F. cylindrus* (−0.283; Table S2). Unexpectedly, Ti also shows significant negative correlations with [HBI:3] (−0.218; Table S2) and positive correlations with the [HBI:2] (0.372; Table S2).

**BGD**

doi:10.5194/bg-2015-610

**Sedimentary response to sea ice and atmospheric variability**

P. Campagne et al.

Interactive Discussion

Variations in the relative abundances of Zirconium (Zr) and Rubidium (Rb) content provide an estimate of the variations in sediment grain size, where Zr represents the coarsest sediment fraction and Rb the finest (Dypvik et al., 2001). In the Weddell Sea and in the Mertz Depression, recent studies have proposed the Zr / Rb ratio as an indicator for enhanced bottom water formation and increased (coarser) sediment transport due and intensification of convective current during periods of intense sea ice formation and brine rejection (Sprenk et al., 2014; Campagne et al., 2015). Zr / Rb is located on F1-&F2+ axes (Fig. 4), and displays highly significant negative correlations with some open water taxa, e.g. *P. glacialis* (−0.193; Table S2), *T. antarctica* (−0,188; Table S2) and *F. kerguelensis* (−0,176; Table S2), coherent with the use of the ratio.

Other open water related proxies, the *Rhizosolenia* gp. (*Rhizolenia* spp. and *Proboscia* spp.), the *Thalassiothrix* gp. (*Thalassiotrix antarctica* and *Trichotoxon reinboldii*) and *F. rhombica*, are discussed in Note S2.

### 4.1.3 Surface water stratification and wind conditions related proxies

Vegetative cells of *Chaetoceros Hyalochaete* are often observed in high abundances within the surrounding pack ice waters (Gleitz et al., 1998) and are abundant in Adélie Land surface waters (Beans et al., 2008). High nutrient content in surface waters, surface water stratification by sea ice meltwater and stabilization by low wind intensity appear to be the most important factors for the development of *C. Hyalochaete* blooms (Leventer, 1991, 1998). It has been suggested that resting spore formation occurs when the bloom begins to decline as a response to decreasing nutrient levels, low light levels during vertical mixing of the water column or as a result of reduced seasonal insolation during the autumn (Hollibaugh et al., 1981). Nutrient depletion in highly stratified surface waters, as a result of both meltwater input and thermal warming (Leventer, 1991), is probably the main trigger for spore formation. Highest sedimentary abundances of CRS (> 80 %) occur in the Antarctic Peninsula with ∼ 7 months yr$^{-1}$ sea ice cover (Armand et al., 2005). In Holocene sediment, high relative abundances of CRS are commonly used to track high productivity events and strongly stratified sur-

**BGD**

doi:10.5194/bg-2015-610

**Sedimentary response to sea ice and atmospheric variability**

P. Campagne et al.

face waters at the receding sea-ice edge (Leventer et al., 1996; Denis et al., 2006; Leventer et al., 2006). *Chaetoceros Hyalochaete* vegetative cells were here counted with CRS, as they are generally present in low abundance in Antarctic sediments. CRS is significantly located on the F1-&F2+ area in our PCA, strongly separated from other diatom species (Fig. 4). CRS display numerous negative significant correlation with both sea ice associated and open ocean associated species (e.g. *F. curta*, *F. cylindrus*, *F. rhombica*, *F. kerguelensis, Chaetoceros cryophilum;* Table S2).

Other wind and stratification condition related proxies such as the *Chaetoceros Phaeoceros* (*Chaetoceros dichaeta and C.cryophilum*), the Benthic gp. (*Cocconeis* spp., *Grammatophora* spp., *Trachyneis* spp., *Licmophora* spp., *Melosira sol* and *M. adelia*, *Achnantes brevipes*, *Amphora* spp., *Diploneis* spp., *Pseudogomphonema* spp., *O. weissflogii* and *Corethron* spp. are discussed in Note S2.

## 4.2   Relationship between sedimentary proxies and instrumental data

Environmental parameters were averaged over the ice-free season (from November to March) to allow a direct comparison of the impact of atmospheric and sea surface conditions on sedimentary response, and by inference, biological surface response. Although the standardization partially hid the variability of our sedimentary signals (Fig. S2), the main characteristics are preserved and allow for interannual comparisons. Results are summarized and presented in Table 1. Hereafter, we use the main diatom species and species groups along with geochemical proxies identified in the previous PCA (Fig. 4).

### 4.2.1   Axis interpretation

On both PCA (Fig. 5) enhanced sea ice concentrations (SIC) and late sea ice retreat are located on F1-, along with late sea ice advance, which is insignificant. In contrast, a longer ice-free season is significantly located on F1+ (Fig. 5). F1 axis, which represents the highest variance of 18.09 % (Fig. 6), represents sea ice condi-

Discussion Paper | Discussion Paper | Discussion Paper | Discussion Paper |

**BGD**

doi:10.5194/bg-2015-610

**Sedimentary response to sea ice and atmospheric variability**

P. Campagne et al.

tions from November to March. As expected, the length of the ice-free season and SIC are strongly linked with sea ice cover dynamics. Pearson correlations indicate that early sea ice retreat and delayed sea ice advance are associated with decreasing SIC and a longer ice-free season, and *vice versa* (Table S3 in the Supplement). The easterly wind direction is significant on F1+ while the southerly wind component is significantly located on F1- (Fig. 5a; Table S3), indicating that sea ice conditions off Adélie Land respond to the dominant wind field. Indeed, our data indicate that more easterly winds are associated to both earlier sea ice retreat and earlier sea ice advance, whereas more southerly winds are linked to later sea ice retreat and later sea ice advance (Table S3), in line with Sect. 2.5 (Cases B and C respectively). The length of the ice-free season does not show any relationship with atmospheric parameters but mainly depends on the timing of sea ice retreat (Table S3), whereby earlier opening would lead to longer open season.

F2 axis accounts for 16 % of the total variance (Fig. 6). Westerly and northerly wind directions are significantly located on F2- and F2+, respectively, with very low scores on F1 (Fig. 5a; Table S3). Westerly and northerly winds further display high positive relationships with the timing of the closing date, but show no links with the opening date. They also show negative relationships with the wind speed (Fig. 5b; Table S3). These results indicate that westerly and northerly wind fields generate delayed sea ice advance in autumn. The westerly wind component is associated with southerly winds and low temperatures (Table S3). Increasing wind direction (southerly to westerly winds) or weak onshore (northerly winds) circulation is of cyclonic origin in our study area (Sect. 2.5, Case A). This result suggests that storm forcing on sea ice conditions are represented by the F2 axis. Additionally, the wind speed parameter seems to be closely associated to the F5 axis (Fig. 5b) and represents a lower contribution of 7.65 % (Fig. 6).

These results agree well with the PCA between seasonal atmospheric forcing and sea ice conditions over the core site depicted in Sect. 2.5, and with several studies at larger spatial scales (Massom et al., 2003, 2009; Smith et al., 2011; Wang et al.,

**BGD**

doi:10.5194/bg-2015-610

**Sedimentary response to sea ice and atmospheric variability**

P. Campagne et al.

2014; Campagne et al., 2015) which argue that wind conditions, and particularly wind direction, exert a strong impact on sea surface conditions off Adélie Land.

### 4.2.2 Sedimentary response

**Westerly winds increase spring sea ice conditions (Case A; Sect. 2.5)**

A dominant westerly wind component originating from enhanced cyclone activity conducts to lower temperatures and wind speed in our data (Table S3). Westerly winds have been observed to promote pack ice or thinner sea ice lasting longer in spring (see Sect. 2.5) and delayed sea ice advance in autumn in our study area (Table S3). Under such conditions, the sedimentary response shown by the PCA indicates increased an abundance of [HBI:2]. Indeed, [HBI:2]/[HBI:3] (0.398; Table S3) is significantly correlated with westerly winds and presents a strong negative relationship with low temperature in our data (Fig. 5). Similar environmental relationships are observed for [HBI:2] (Fig. 5), agreeing with the environmental interpretation of [HBI:2] as a sea ice indicator.

**Northerly winds increase spring sea ice conditions (Cases A or B; Sect. 2.5)**

At the Antarctic scale, northerly winds promote early spring sea ice retreat and late autumn sea ice advance (Stammerjohn et al., 2008). However, northerly winds in our study area have been observed to decrease northward transport of sea ice (Holland et al., 2012). Our PCA results (Fig. 5) indicate that the northerly wind component is associated with lower wind speeds conducive to delayed sea ice advance at the core site (see Sect. 4.2.1), but relationships with SIC and sea ice retreat date are not clear in our data. We propose that northerly winds have two opposite impacts on sea ice conditions off Adélie Land. First, they tend to increase spring sea ice presence in the area, likely by pushing the offshore pack ice toward the coast. Secondly, they contribute to increased summer melting and limit ice growth in autumn by transporting warm air from the north and by enhancing the swell. Under such environmental conditions, the sedimentary

**BGD**

doi:10.5194/bg-2015-610

**Sedimentary response to sea ice and atmospheric variability**

P. Campagne et al.

response shown by the PCA indicates concomitant increasing abundances of the *Rhizosolenia* gp., [HBI:3], *F. cylindrus* and *E. antarctica,* that have high scores on F2+ (Fig. 5a). The concomitant presence of sea ice related proxies and open ocean proxies indicate a marked seasonal cycle. *Fragilariopsis cylindrus* displays a strong association with high temperature in our study area, but present a non-significant positive relationship with SIC (Table S3). These results may indicate that this species would respond to heavier pack ice conditions in spring and subsequent melting. Similarly, *E. antarctica* is strongly related to northerly winds through later sea ice retreat and shorter ice free season in our data (Fig. 5; Table S3). The presence of [HBI:3], a marker of open water related productivity, along with *Rhizosolenia* gp., an indicator of warmer conditions and of a long diatom productivity season (see Sect. 4.1.1), would support the presence of stable sea surface conditions due to longer melting conditions during spring-summer and with enhanced open water conditions in autumn.

**Easterly winds induce open conditions (Case B; Sect. 2.5)**

A dominant easterly wind component yields to earlier sea ice retreat and earlier sea ice advance, less SIC from November to March, and in some extent to a longer ice free season in the study area. Under such environmental conditions, the PCA indicates that the sedimentary response consists of higher abundances of several proxies strongly linked to F1+, *Thalassiotrix* gp., *T. antarctica*, *P. glacialis*, *F. kerguelensis*, *Fragilariopsis* summer gp., the Benthic gp., [HBI:3] and Ti (red shaded area on Fig. 5). As they are closely positioned on F1+, these diatom species/gp. and geochemical proxies display a strong positive intercorrelation (Table S3) as well as being correlated with the easterly wind component. The Ti signal is also highly linked to the northerly wind component (Table S3), suggesting that this proxy is tied to onshore/westward wind circulation and probably responds to longer ice–free season. *Thalassiotrix* gp., *T. antarctica*, *P. glacialis*, *F. kerguelensis* and Ti are strongly linked to low SIC over the ice free season in our study area (Table S3). Similarly, [HBI:3], *P. glacialis* (−0.431; Table S3)*, F. kerguelensis* (−0.360) and *T. antarctica* are strongly associated with earlier open water

**[BGD]**

doi:10.5194/bg-2015-610

**Sedimentary response to sea ice and atmospheric variability**

P. Campagne et al.

conditions, and *F. kerguelensis* is linked to a longer ice-free season. Open water gp., *C. cryophilum*, *F. kerguelensis,* Ti and [HBI:3] are highly associated with ice free seasons characterized by weak winds, suggesting that these species prefer more stable environments. However some differences also appear in the environmental response of

5 these proxies. While the *Thalassiotrix* gp. is strongly linked with more open conditions in autumn, *P. glacialis* and *T. antarctica* are, contrastingly, associated with an earlier sea ice advance and lower temperature (Fig. 5; Table S3). These results agree well with ecological preferences of *P. glacialis* and *T. antarctica,* which are considered as biological indicators of early sea ice conditions in autumn. However, in our study *P. glacialis*

is linked to a longer ice free season, contrary to *T. antarctica* (Fig. 5; Table S3). These results contradict paleoecological inferences on these species whereby *P. glacialis* has been associated with areas experiencing a slightly longer annual sea ice cover relative to *T. antarctica* (Armand et al., 2005; Pike et al., 2009), maybe because these studies did not separate the two varieties of *T. antarctica*.

Our results are generally in line with the known ecology of this geochemical and diatom assemblage (Sect. 4.1.1). Several of these proxies are associated with enhanced open conditions during the ice free season in the area. Indeed, increasing abundances of *T. antarctica*, *F. kerguelensis, P. glacialis*, large centric diatoms that constitute the Open water gp., [HBI:3] and Ti have been attributed to more open conditions in the

Mertz Glacier Polynya area, during periods with reinforced easterly wind conditions (Campagne et al., 2015). This sedimentary assemblage that is associated to an easterly wind promoting open water conditions may therefore reflect the development of the DDU polynya (Arrigo et al., 2003) during the early spring-summer period in the study area.

**Katabatic winds and coastal polynya (Case C; Sect. 2.5)**

A dominant southerly wind component is conducive to a delayed spring sea ice retreat and late autumn freezing, higher SIC from November to March, and to some extent, a shortened ice free season in the study area (Sect. 4.2.1). Strong southerly

**[BGD]**

doi:10.5194/bg-2015-610

**Sedimentary response to sea ice and atmospheric variability**

P. Campagne et al.



winds (katabatic) produce open water conditions near the coast but not necessarily at the core site. Southerly winds allow more fast ice to be transported northward of the core site (Sect. 2.5) by restoring the eastward drift of the pack ice. Under such environmental conditions, the sedimentary response shown by the PCA indicates increasing abundances of CRS and Zr/Rb that are strongly linked to F1- (Fig. 5a). Their response to katabatic wind events is probably through the opening of the coastal polynya (Sect. 2.5). In addition, although CRS shows no relation with SIC parameters, this group is associated to a strong wind speed (0,364; Table S3) over the ice-free season, in agreement with the environmental interpretation of the species (Sect. 4.1). In contrast, Zr/Rb is closely associated with increasing SIC over the ice-free season in addition to a delayed sea ice retreat (Table S3).

Enhanced polynya activity along the coast has been shown to coincide with deep mixed layer and high Chl *a* levels (Vaillancourt et al., 2003; Riaux Gobin et al., 2013). Stronger vertical mixing may increase nutrient availability; favouring the rapid development of *Hyalochaete* spp. Spore formation may occur through deep mixing of vegetative cells under the photic zone or nutrient depletion when winds weaken. Northward lateral advection of surface production from the coast to the study area may explain the occurrence of CRS along with more sea ice at the core site. Continuous sea ice formation in the coastal polynya may have favoured brine production and subsequently, stronger bottom currents, leading to the observed Zr/Rb values.

## 4.3 Which proxies and environmental interpretations for long term reconstructions

We here discuss the relevance of our results as proxies for long term reconstructions, by commenting on their co-occurrence with environmental parameters over the last 40 years.

**BGD**

doi:10.5194/bg-2015-610

**Sedimentary response to sea ice and atmospheric variability**

P. Campagne et al.

### 4.3.1  Sea ice conditions

Highest abundances of *Fragilariopsis cylindrus* cells in sediments are observed during the ~ 1972–1979, 1983–1984, 1996–1998 and 2002–2003 CE periods (Fig. 7). At the same time, analysis of the satellite data reveals the presence of heavier sea ice conditions in the area during spring, summer and autumn in the 1979–1981, 1990–1998 and 2001–2004 CE periods (Fig. 8). Although these intervals broadly coincide with those characterized by higher abundances of *F. cylindrus*, ice concentration during the productive season does not satisfactorily explain the variability of species abundance during the last 40 years. Since this species has previously been reported as favouring environments characterized by the presence of sea ice in spring (Kang and Fryxell, 1992; Armand et al., 2005), cell abundances were compared with the timing of the sea ice retreat. Again, both records displayed only a weak (or non-existent) relationship (Fig. 8). Interestingly, a strong relationship is observed between F. cylindrus abundances and wind origins. The increasing occurrence of relatively weak northerly winds (N / W ratio > 0.9), bringing warmer water masses and somewhat advecting pack ice into the area during 1971–1975, 1979–1980, 1984 and 1992 and in the 1996–2001 CE period (Fig. 8), would contribute to the stratification of surface waters with large amounts of sea ice meltwater that have also been identified as promoting the development of this species. The [HBI:2] concentrations increased in 1971–1978, ~ 1982, 1985–1986, ~ 1989, 1995–2002, and since 2006 CE (Fig. 7). As observed for F. cylindrus, [HBI:2] concentrations do not closely follow SICs and the sea ice dynamics at interannual scale over the last 40 years. Rather, high concentrations of this biomarker co-occur with more westerly winds, as the N / W ratio is low between 1975–1979, 1981–1982, 1985–1991, 1993–1996, 2000 and since 2002 CE (Fig. 8), suggesting [HBI:2], and sea ice dwelling diatoms, would have greater affinities with pack ice conditions relative to fast ice. Our results therefore attest to the importance of the origin and nature of the sea ice in the sedimentary distribution of sea ice proxies in our study area. Distinction of the two sea ice structures through finer satellite study in the region (e.g. MODIS satellite imagery)

Discussion Paper | Discussion Paper | Discussion Paper | Discussion Paper |

**BGD**

doi:10.5194/bg-2015-610

**Sedimentary response to sea ice and atmospheric variability**

P. Campagne et al.

would help to assess and better constrain such relationships. Although [HBI:2] and *F. cylindrus* are both associated with increasing pack ice in our study area, the biomarker record lags the diatom record by ∼ 2 years in core DTCI2010. This may suggest that one species (*F. cylindrus*) responds better to warmer (melted) conditions relative to the other species ([HBI:2]). However, time series studies on the formation, export and burial of HBIs and diatoms are necessary to fully understand this decoupling. We note that the two proxies present similar trends at sub-decadal scale, supporting their use as complementary sea ice indicators for paleoclimate studies.

The *T. antarctica/Porosira* gp. ratio increased slightly over the 1976–1978, 1982–1985 and ∼ 1995–1996 CE periods, reaching highest values between 2000–2010 CE (Fig. 7). As already observed for previous sea ice related proxies, the distribution of autumnal bloom species does not closely follow SICs variability in our study area. However, the large *T. antarctica/Porosira* gp. increase over the last decade coincides with the earliest sea ice advance in the last 30 years, occurring in March, which resulted from the prevalence of easterly winds (Fig. 8). A significant shift to more easterly winds in Adélie Land was concomitant to a southward migration of the mid-latitude Westerlies and to positive Southern Annular Mode (SAM) values since 1960 CE (Campagne et al., 2015), the latter being the principal mode of atmospheric variability over the Southern Ocean (Thompson and Solomon, 2002). Enhanced easterly winds have been observed to weaken the northward transport of sea ice in the region, increasing sea ice conditions (Massom et al., 2003, 2009). This suggests that *T. antarctica* and *Porosira* spp. are likely good indicators for autumnal sea ice dynamics forced by atmospheric variability.

### 4.3.2 Open ocean conditions

At the opposite of the sea ice related proxies, open ocean proxies seem to follow sea ice conditions in our study area relatively well, regardless origin, structure and nature. Indeed, the Open Water gp. and *F. kerguelensis* assemblages display similar patterns, increasing in 1972–1977, in 1980–1986, in 1995–2001 and at ∼ 2007 CE (Fig. 7),

Discussion Paper | Discussion Paper | Discussion Paper | Discussion Paper |

**BGD**

doi:10.5194/bg-2015-610

**Sedimentary response to sea ice and atmospheric variability**

P. Campagne et al.

which coincides with periods of low SICs, earlier sea ice retreat and enhanced ice free season (Fig. 8). Similarly, [HBI:3] exhibits high values in ∼ 1973, ∼ 1978, 1983–1985 CE, and moderate increase in 1995–2001 CE (Fig. 7), supporting a strong relationship with ice- free conditions. The maximum occurrence of [HBI:3] between the mid 1970s to the mid 1980s coincides well with positive temperature anomalies and lower wind speed anomalies at that time (Fig. 8), arguing for a strong relationship between biomarker development and surface conditions forced by atmospheric variability. The open ocean proxies discussed above further present a similar long term trend in core DTCI2010, supporting their use as indicators of the lengthening of ice free conditions in paleoclimate studies. Additionally, these proxies are relatively coherent with the Ti increasing values in 1975–1978, 1985–1987, 1996–2001 and ∼ 2008–2010 CE (Fig. 7). However, highest Ti values occurred in 1970–1972 CE, concomitant to a slight increase of open ocean proxies and no particular warming trend in monitored temperatures (Fig. 8). This suggests that Ti content can be impacted by rapid terrigenous events that are independent of the length of the ice-free season (Fig. 8). Therefore, downcore differences between Ti and open ocean proxies may help in identifying episodic events such as local glacial discharge or ice rafted material derived from the Mertz Glacier Tongue basal melting (Maddison et al., 2006; Dinniman et al., 2012; Campagne et al., 2015).

### 4.3.3 Polynya activity

*Chaetoceros* RS and Zr / Rb exhibit contrasting trends during the 1970s to the mid 1980s, with decreasing (increasing) CRS (Zr / Rb) values during the 1973–1976 CE and the ∼ 1977–1978 CE periods, and are then relatively in phase with concomitant increasing values in 1984/85–1987, and in 1989–1995 CE (Fig. 7). Zr / Rb follows SIC variability relatively well whereas CRS relative abundances follow spring sea ice dynamics (Fig. 8). It is also worth noting that both proxies coincide well with prevalence of southerly winds in our study area between 1983–1996 CE (Fig. 8). Such a change in their relationships during the 1970–1980s may highlight a decoupling between polynya

**BGD**

doi:10.5194/bg-2015-610

**Sedimentary response to sea ice and atmospheric variability**

P. Campagne et al.

Discussion Paper | Discussion Paper | Discussion Paper | Discussion Paper |

activity and katabatic winds. While CRS are strongly linked to katabatic wind pulses and rapid stratification/mixing events, polynya activity (and bottom water velocity) may respond to windy conditions, whatever the wind origin. In Adélie Land, maximum wind speeds have been found for prevalent south to easterly winds (Sect. 2.5). This, com-
bined with increases of Zr / Rb between 1970–1985 CE and highest open ocean prox- ies values, suggests a balance between easterly and southerly winds over the ice free season during this period, promoting early spring polynya activity and summer open conditions. Subsequently, strong and persistent easterly winds, along with in- creasing onshore wind circulation, may have softened polynya activity since the mid
1990s (Figs. 7 and 8).

## 5  Conclusions

Investigation of annual to interannual relationships between diatom communities, di- atom specific biomarkers and major element abundances in connection with meteo- rological parameters show that the relevance and use of some proxies (1) may be
characteristic of the study area (e.g. CRS assemblage), or (2) differ slightly from pre- vious long term studies (e.g. *F. cylindrus*). Indeed at such a fine scale, the distribution of the sea ice related proxies in sediment highlights complex relationships of the biota with sea ice concentration, sea ice dynamics, sea ice cover structure and origin, con- strained by the wind pattern. However, we note that the complex relationship between
winds and sea ice should be alleviated at longer time scales if larger atmospheric and ocean temperatures changes become more preponderant. Open ocean related proxies seem to primarily respond to the lengthening of the growing season in our study area, agreeing well with previously published regional to large scale studies. Polynya activity may be inferred from variations in the Zr / Rb ratio values. Monitoring the sedimentary
signal formation in surface water, through the water column and its export and burial in deep-sea sediments is necessary to understand the local behaviour of the proxies commonly constrained via synoptic studies. Other high-resolution and longer timescale

Discussion Paper | Discussion Paper | Discussion Paper | Discussion Paper | Discussion Paper |

**BGD**

doi:10.5194/bg-2015-610

**Sedimentary response to sea ice and atmospheric variability**

P. Campagne et al.

reconstructions are required to refine our understanding of the ice–ocean–atmosphere interactions and system feedbacks.

**The Supplement related to this article is available online at doi:10.5194/bgd-13-1-2016-supplement.**

*Author contributions.* X. Crosta and G. Massé designed the study and P. Campagne carried it out. O. Ther performed diatom and XRF analyses; P. Campagne performed diatom sensus counts and PCA analyses; M-N. Houssais extracted daily sea ice concentrations; S. Schmidt performed $^{210}$Pb analyses and developed the age model of the core; P. Campagne prepared the manuscript with contributions from all co-authors.

*Acknowledgements.* This research was funded by the ERC StG ICEPROXY project (203441), the ANR CLIMICE project and FP7 Past4Future project (243908). The CNRS (Centre National de la Recherche Scientifique) and the FRQNT (Fonds de recherche du Québec – Nature et technologies) provided the student fellowship. The French Polar Institute provided logistical support for sediment and data collection (IPEV projects 452 & 1010). This is ESF PolarClimate HOLOCLIP contribution no. 24 and Past4Future contribution no. 83. The authors thank Debra Christiansen-Stowe and Julie Sansoulet for logistical and administrative assistance.

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

**Table 1.** Summary of the relationships between sedimentary proxies and environmental conditions.

| Proxy/Group identified in core DTCI 2010 | Known ecology (literature) | Environmental relationships off Adélie land |
|---|---|---|
| *F. cylindrus* | Spring sea ice covered/sea ice stratified waters | Onshore winds favour warm conditions and melted pack ice in spring |
| [HBI : 2], [HBI : 2]/[HBI : 3] | Spring sea ice environment | Westerly winds increase spring compacted (cold conditions) pack ice conditions in spring |
| Banquisia gp. (*N. directa N. glaciei, Synedra* spp., *Ephemera* spp.) | Spring sea ice conditions | No clear pattern |
| *F. obliquecostata* | Surface melt pools, sea ice covered waters | No clear pattern |
| *E. antarctica* | Ubiquist, open conditions | Onshore winds increase spring-summer sea ice (short ice free season, delayed sea ice retreat) |
| *Fragilariopsis* summer gp. (*F. ritscheri, F. sublinearis*) | Summer sea-ice edge environment, melted waters | No clear pattern |
| *T. antarctica* | Open water conditions in summer-autumn, slush and wave-exposed shore ice environment | Easterly winds induce summer open water conditions but icy autumn conditions (early sea ice advance) |
| *Porosira* gp. (*P. glacialis, P. pseudodenticulata*) | Relative open water conditions in summer, slush and wave-exposed shore ice environment, icy autumn and early sea ice advance | Easterly wind induce summer-autumn open water conditions (long ice free season, relatively early sea ice advance) |
| *F. kerguelensis* | Open water-ice free conditions during summer | Easterly winds induce summer open water conditions (earlier sea ice retreat, long ice free season) |
| Open water gp. (*T. lentiginosa, T. oliverana, T. trifulta, T. gracilis, T. tumida, Coscinodiscus* spp., *A. actinochilus, S. microtrias*) | Open water-ice free conditions during summer | Easterly winds induce summer open conditions (earlier sea ice retreat, long ice free season) |
| [HBI : 3] | Open water-ice free conditions during summer | Easterly-onshore winds induced summer open water conditions (early sea ice retreat) |
| *Rhizosolenia* gp. (*Rhizolenia* spp., *R. antennata semispina, Proboscia* spp., *P. truncata* and *P. inermis*) | Open conditions in autumn, highly stable and nutrient poor waters | Onshore winds induce stable surface waters, no clear trend with sea ice conditions |
| *Thalassiotrix* gp. (*Tx. antarctica, T. reinboldii*) | Open conditions in autumn, highly stable and nutrient poor water column | Easterly winds induce summer-autumn open water conditions (low SIC, long ice free season, early sea ice retreat and later sea ice advance) |
| *F. rhombica* | Spring-summer ice free-less icy conditions | Warm conditions in spring-summer, delayed ice free season (late sea ice retreat and late sea ice advance) |
| Ti | Summer open conditions, melted conditions | Onshore winds favour warm conditions, increase melting of sea ice in spring, open water conditions in summer |
| Zr / Rb | Polynya environment | Katabatic winds induce coastal polynya and sea ice formation (high SIC) |
| CRS (*Chaetoceros Hyalochaetes* spp.) | Rapid changes in stratification, decreasing nutrient levels | Katabatic wind pulses induce coastal polynya or turbulent surface layer events (high wind speed) |
| *Phaeoceros* gp. (*C. Phaeoceros* spp., *C. atlantica, C. dichaeta*) | Open water environment | No clear pattern |
| Benthic gp. (*Cocconeis* spp., *Grammatophora* spp., *Trachyneis* spp., *Licmophora* spp., *Melosira solenia, Achnantes brevipes, Amphora* spp., *Diploneis* spp., *Melosira adelia, Pseudogomphonema* spp., *O. weissflogii*) | Spring wind (storm) induced mixing conditions | No clear pattern |
| *Corethron* gp. (*C. criophilum, C. pennatum*) | Open ocean conditions, surface mixed waters | No clear pattern |

Discussion Paper | Discussion Paper | Discussion Paper | Discussion Paper |

**BGD**

doi:10.5194/bg-2015-610

**Sedimentary response to sea ice and atmospheric variability**

P. Campagne et al.

Discussion Paper | Discussion Paper | Discussion Paper | Discussion Paper

**BGD**

doi:10.5194/bg-2015-610

**Sedimentary response to sea ice and atmospheric variability**

P. Campagne et al.

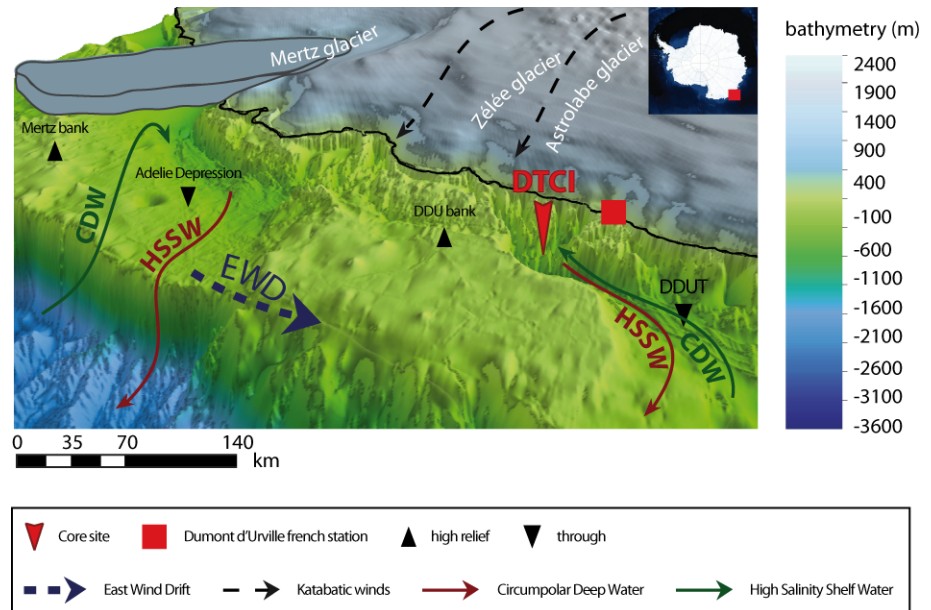

**Figure 1.** Study area. Map of the study area showing the location of sediment core DTCI 2010 in the Dumont D'Urville Trough (DDUT), the main glacial and topographic features, and the principal water masses.

**BGD**

doi:10.5194/bg-2015-610

**Sedimentary response to sea ice and atmospheric variability**

P. Campagne et al.



**Figure 2.** PCA seasonal data of weather forecast parameters. PCA applied to weather forecast and satellite data, that were previously seasonally averaged.

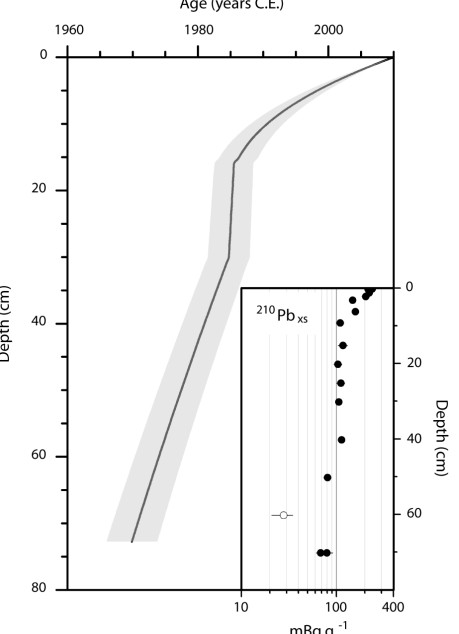

**Figure 3.** DTCI 2010 chronology, based on $^{210}$Pb excess ($^{210}$Pb$_{xs}$) and associated age-model errors (grey area). The inset corresponds to the downcore profile of $^{210}$Pb$_{xs}$ (error bars correspond to 1 SD).

Discussion Paper | Discussion Paper | Discussion Paper | Discussion Paper | Discussion Paper |

**BGD**

doi:10.5194/bg-2015-610

**Sedimentary response to sea ice and atmospheric variability**

P. Campagne et al.





**BGD**

doi:10.5194/bg-2015-610

**Sedimentary response to sea ice and atmospheric variability**

P. Campagne et al.

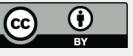

**Figure 4.** PCA applied to sedimentary raw data from the DTCI 2010 core. Shaded areas represent diatom clusters, based on significant correlation between species (Table S2). Abundant species (relative abundance > 2 %) are written in black, unrepresentative species (relative abundance < 2 %) are written in grey.

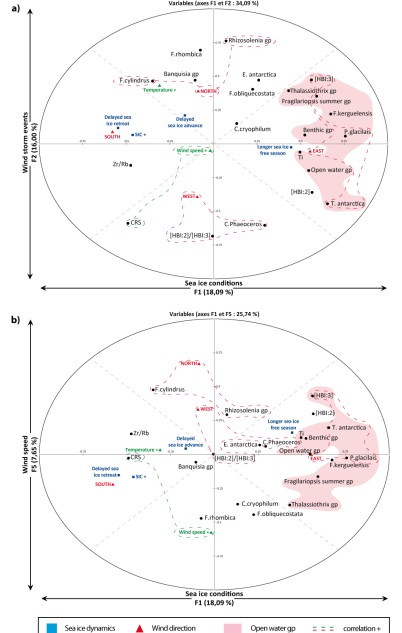

**BGD**

doi:10.5194/bg-2015-610

**Sedimentary response to sea ice and atmospheric variability**

P. Campagne et al.

Title Page

Abstract | Introduction

Conclusions | References

Tables | Figures

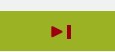 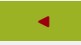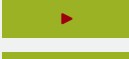

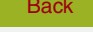

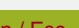



**Figure 5.** PCA applied to standardized sedimentary data from the DTCI2010 core and meteorological parameters. The yearly standardized sedimentary data represent the ice-free season of the related year. Weather forecast/satellite data were averaged between November and March. **(a)** F1 axis represents the sea ice conditions linked to the predominant Easterly and Southerly winds. F2 axis represents the secondary wind directions in the study area, Northerly and Westerly winds. **(b)** F1 axis represents the sea ice conditions linked to Easterly and Southerly winds. F5 axis represents the wind speed.

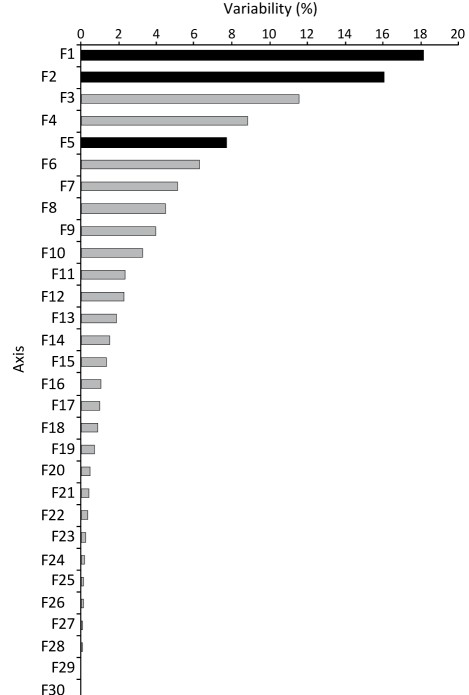

**Figure 6.** Axis contribution to the total variability, from PCA applied to standardized sedimentary and meteorological parameters. Selected axes for PCA are in black.

Discussion Paper | Discussion Paper | Discussion Paper | Discussion Paper | Discussion Paper |

**BGD**

doi:10.5194/bg-2015-610

**Sedimentary response to sea ice and atmospheric variability**

P. Campagne et al.

Title Page

Abstract · Introduction

Conclusions · References

Tables · Figures

|◄ · ►|

◄ · ►

Back · Close



Discussion Paper | Discussion Paper | Discussion Paper | Discussion Paper

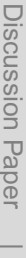

**BGD**

doi:10.5194/bg-2015-610

**Sedimentary response to sea ice and atmospheric variability**

P. Campagne et al.



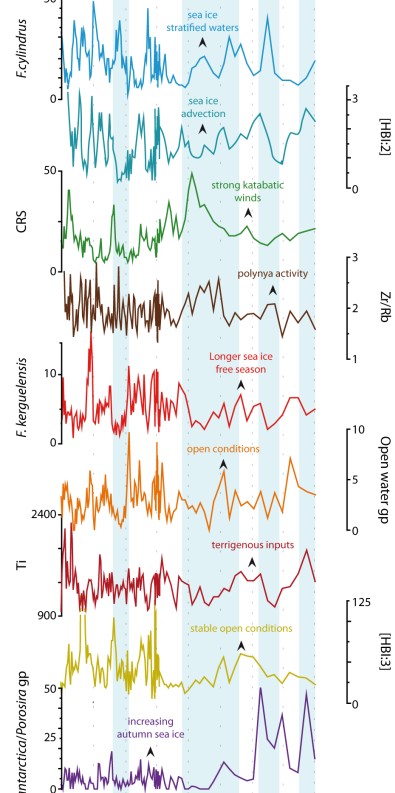

**Figure 7.** Raw sedimentary records from DTCI 2010 interface core over the 1970–2010 period. The blue shading indicates periods marked by increasing sea ice concentration at the core site.

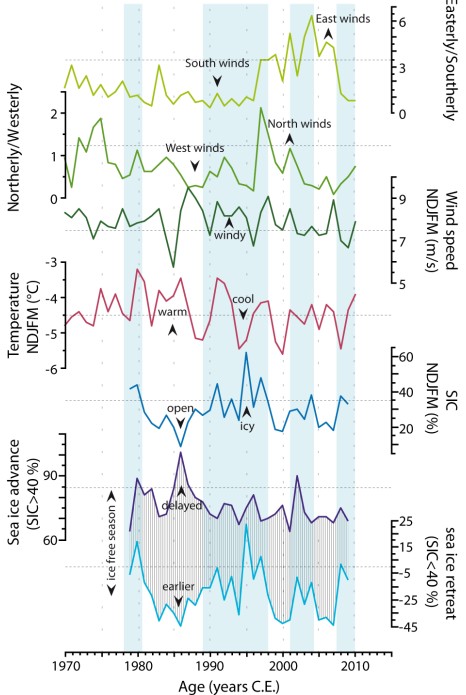

**Figure 8.** Meteorological parameters and climate index over the 1970–2010 period. Daily meteorological parameters were averaged over the November to March period. Blue shaded areas mark increases of SIC.

Discussion Paper | Discussion Paper | Discussion Paper | Discussion Paper

**BGD**

doi:10.5194/bg-2015-610

**Sedimentary response to sea ice and atmospheric variability**

P. Campagne et al.

