# Peer review of "Sedimentary response to sea ice and atmospheric variability over the instrumental period off Adélie Land, East Antarctica"

_Biogeosciences, 2015_

## Referee Comment (RC1) · Anonymous Referee #1 · 26 Feb 2016

General Comments

Diatom has been used as a proxy of paleoenvironmental information based on large-scale diatom ecological studies, whereas little is known about local-scale diatom ecology. The authors attempted to solve this issue namely understanding the relationship between diatom communities and local environmental conditions using the principal component analyses (PCA) and the Pearson correlation test.

Although I appreciate authors efforts, the criteria that authors set for categorizing environmental influences seems not fully quantitative. I think the paper requires restructuring since some sections are lacking information, whereas other sections are lengthy description that may be put into the supplement. Therefore I believe the current version

of the manuscript is not satisfied the requirements to be published in the journal.

I describe specific comments below:

Introduction

Authors are needed to review previous studies on similar topics and discuss the problems remaining in the topic discussing in the paper. Some information is available in the text but in more comprehensive manner is requested.

Environmental settings

Authors should provide the detailed description of the quantitative criterion. Also raw data of meteorological data need to be provided so that readers can evaluate them.

Material and methods

Information is lacking how the measurements of 210Pb chronology and biomarker analyses are conducted. Again raw data of 210Pb measurements and diatom assemblages are required either in the main text or in the Supplement.

A question is also raised how to standardize different sort of data (diatom, proxies and meteorological parameters) in PCA? It seems that there is no need to do PCA using 67 diatom species data. Authors should conduct PCA using only abundant species.

Results and discussion

Some re-organization of the section is required. The section 4.1 and 4.2 are too long, a part of these sections can be moved to the Supplement. It is appropriate to discuss the axis interpretation (subsection 4.2.1) first then discuss the relationship between sedimentary proxies (section 4.1). A part of the second paragraph of the section 4.1.1 should be moved to the Methods section.

The genus of diatom that has been mentioned is spelled out and abbreviate in the following sections.

Figure

The figures would be required for some editing. Sites discussed in the text should be added in the figure 1. Also add the dominant direction of winds. Photographs and stratigraphic columnar sections of core should be presented. Figure 5 is too small to decipher and bigger figure is needed. Authors should describe details of figures such as the meaning of black arrows in the Figure 7? What does dashed line in the Figure 8 mean? Moreover, authors should write environmental information (e.g. east winds, south winds) outside the graph to make the figure clear and concise.

Supplement

Some re-organization of the supplement text is required with "Results and Discussion" section. Authors should adjust the table size appropriate for publication. Table S2 is too large (continued 7 pages!!), and Table S3 is illegible for the small size.

Page 13, Line26: tri unsaturated -> tri-unsaturated

Page 14, Line 3: co occurrence -> co-occurrence

Page 21, Line 12: Corethron spp. are -> Corethron spp.) are

Figure 7: Add the unit of y-axis.

Figure S2: Add the unit of x-axis and y-axis.

Table S1: Add the description of each meteorological parameters to the table caption.

---

## Referee Comment (RC2) · L. Armand (Referee) · 31 Mar 2016

General Comments. This paper focuses on the understanding the diatom, elemental and isoprenoid records preserved in an ∼40 year old, annually laminated, interface core from the Durmont D'Urville Trough, Adelie Land, East Antarctica. The study attempts to relate the changes in down core records of the preserved biological and chemical records to both prevailing regional atmospheric and sea-ice conditions for the same time period assuming the laminations are linked.

Although the paper makes a significant advance to our understanding of the environmental conditions of the Adélie –George V Land coastal region as is recorded in a 40 year time series, the paper requires a major refocusing or dividing in to two papers so

that story is more succinctly and clearly delivered. At present it is very hard to read through and follow how the main aims are delivered on. It needs to be significantly reorganised and cut back, which I strongly encourage all the co-authors to assist the first author in undertaking.

The results need to be extracted into their own section and subsections and therefore separated from the heavy discussion sections in which they are currently hidden. Much of the ecological discussion should be cut back and either placed in if relevant to the current understanding of climate interpretations in the Introduction or used to support/provide contrast to the results in a discussion section. For example: qualifications of species groupings and/or resting spores/phases that have been applied prior to analysis should be found in the methods section (3.2) and not in the discussion.

Terminology needs careful attention (for example seasonal sea-ice zone = the area between the winter and summer sea-ice extremes; whilst marginal ice zone = the low sea ice concentration typically found at/near the ice edge and can be either quite diffuse and expansive (and hundreds of kilometres wide) or quite narrow and less than a few kilometres, usually depending on the winds). The use of a hyphen is required for sea ice when used as an adjective (e.g. sea-ice edge, sea-ice concentrations, sea-ice extent) but is not used for the noun (e.g. Antarctic sea ice, polar sea ice). Hyphens should also be used for sea-surface temperature (and salinity).

Adherence to basic protocols in the identification of diatom species in the text should be followed, where the first time that the species is mentioned in the text it should be written in full (Fragilariopsis cylindrus) and then the genus name can be abbreviated thereafter (F. cylindrus), with exception to the start of a sentence.

Someone will need to check the English to capture minor errors that appear throughout the paper and to add commas where required. I have sent an annotated copy of the paper to the author for this purpose.

There are a lot of acronyms – this might want to be reconsidered, as to how necessary

some are.

Overall, my recommendation is rejection but with major encouragement to reconsider resubmission of two related papers, one building on the other. If the authors don't like that idea, then I have provided some suggestions on how you might want to revise the current structure including all of what you have here.

Specific Comments Abstract: Needs a rewrite to be more specific on the results identified and more direct about what the climatic relationship may be (i.e does a species or laminae relate to a oceanic, sea ice or atmospheric condition?)

1970-2010 = ∼40 years = 72.5cm long CE or AD ?? Editor might need to decide on what they prefer in this journal. Use of contrasting sea ice zone (SIZ vs Marginal IZ) are these zone what you really mean? What are the major elemental "abundances" (concentrations?) that are significant to this study? Can you be more specific? Results are not clearly identified or conveyed- what about the HBI signal? Last sentence should be removed and replaced with a more specific statement on interpretation and impact resolved.

1. Introduction I suggest adding two more short paragraphs (1) on relevant diatom coastal interpretations taken from discussion 4.1 sections and (2) on current HBI interpretations also found in discussion 4.1 sections

2. Environmental Sections 2.1 minor grammatical error (limit→ limits) 2.2. needs tightening up on the English Sentence change suggestions. '. . . originates from brine rejection, during winter sea-ice formation. . .' sea bottom → sea floor? 'Adelie Land Bottom Water is a major contributor to Antarctic Bottom Water'. 2.3 Sentence change suggestions '. . . coinciding with maximum wind speeds...' 2.4 'Sea-ice' conditions This section would benefit significantly by having a figure/images illustrating the core's location in context to the major sea-ice variations described. I would suggest moving the first sentence of this section to follow after the second. 2.5 Please describe more specifically in what way these studies are limited in time and low resolution (in reference to the state-

ment: Most of these studies are either limited in time (refs) or are too low in resolution . . .) What is AWS? There is no lead in from this section to the case studies/scenarios that follow- it is very disjointed. 2.5.1 to 2.5.3. I am not sure all the detail presented in these three subsections is necessary within the paper. You need to decide if these subsections are another study that needs to go in a separate paper altogether that you can refer to, or that you have completed a summary analysis (detailed in Supplementary section 1), which is then categorised by the three condition types that you apply as standard atmospheric conditions in this paper's analysis (potentially this could be summarised within one paragraph), or if the analysis itself is a major part of the study and the Case studies should be in the results section instead. At present it is too much information, and looks like results of a separate study and thus ultimately detracts from the paper's aim. The section needs a rethink in terms of what purpose it is serving in this paper. Some final comments on the case studies : it isn't clear on what time scales the associations have been based 1 year, 5 years, 20 years etc. . ... There are too many acronyms. 3 Materials and Methods 3.1 Should the word (CANBERRA) be in the text? Did you mean the analyses were undertaken at the Australian National University in Canberra, Australia? I suggest writing the text in the past tense.

3.2 There are two different references outlined in two sentences n this section as having been followed (Crosta and Koç 2007 and Crosta et al. 2004), this should be clarified better. The references used in taxonomic identification should have been detailed here (even if in an appendix or supp. material) particularly given that some of the sea-ice taxa are not necessarily found in the Crosta works listed. Details defining the categories, groupings of taxa or use of resting spores (RS) should have been documented here rather than in the discussion sections. It may be worthwhile placing reference to Table 1 in this section given it is a summary of the published diatom ecological data. Remove the third sentence, but retain the NOA61 usage, detail the type of microscope. Indicate where the data is can be accessed (in a proper public data repository for future use) particularly if you do not provide a sup. table with this information with the paper. How can anything be verified from this work? Note here you indicate 70 species

identified and only 25 species are greater than 2% of total relative abundance, later you indicate under PCA there are 65 species that are used many with abundances less than 1%. I would like to see a table summarising the diatom abundance data in the paper at the very least and related to full public data repository table. 3.3. The last statement is not sufficient (details about analytical parameters found elsewhere). Equally, an indication of where the data are available from should be identified here. 3.4 Can you please check all the dates (1978-2012; 1956-2011; 1979-2009) and make it clear what series you are using, or confined to, for the various instrumental data series given you have from Pb dates indicating the core is 40 years old (1970-2010). I found this section very difficult to follow any reasoning, when some years went beyond or started prior to the core's actual record of preserved conditions. Remove the reference to PCA here as you are only describing the data used not the analyses undertaken upon it later. Why was the 40% sea-ice concentration threshold chosen as the difference between sea ice covered and non-covered periods? Is this threshhold one that has been identified by the sea-ice observation community as a valid concentration along the coast of Adélie Land for this sort of open or closed cover? I know that it makes sense out in the offshore SIZ, but wonder if that holds true close to the Adélie Coast. Please provide a reference and/or images in Section 2.4 that help to support your choice. If it was arbitrarily chosen to represent these conditions, as they occur in the offshore SIZ, then just say so. 3.5 Please make sure that you add the two identified points provided as reasons behind the PCA analyses to your introduction aims undertaken to answer your hypothesis. Start a new paragraph with the text beginning with 'In a second PCA. . .' and then rewrite this being clearer about what you mean by 'significant sedimentary data, in terms of population and ecological preferences' and 'interpolated at one year' (did you mean averaged over one year?). Delete the last half of the last sentence or weave that into the Introduction (, to support. . ..).

Should there have been a material and methods section also focused on the elemental analyses undertaken?? Where is this data located publically?

4. Results and discussion. The majority of text in the first part of Section 4 could be used in the Introduction; much is neither results nor discussion. Also 'George V Land' typo fix.

4.1 Should be re-labelled as the RESULTS section. In here you really should place the results of each analysis that has been covered in the order previously delivered in the paper under methods and materials. Tables, Figures and reference to Supplementary metadata should be identified in the relevant results subsections. Having read through the rather hefty sub-sections of section 4. I have two suggestions on how you may want to reconfigure the paper to provide a clearer message. (1) if you are to keep all elements, these are my suggested new headings based on what you have all through section 4.

4.1 Results 4.1.1 Atmospheric Analysis – resulting in three scenarios – if you make it part of the paper's results. 4.1.2. Pb210 dating/age model ?? 4.1.2. Diatom distributions and abundances in the core in general. 4.1.3. PCA analysis of diatoms – clustering results. Diatoms of significance to this study. 4.1.4 HBI Analyses down core 4.1.5 Elemental analyses down core 4.1.6 PCA/Pearson's analyses on diatoms and environmental parameters

4.2Discussion. 4.2.1 Significant diatoms and environmental significance (to sea ice, HBI, Atmospheric, ? elemental) in general. 4.2.2 Scenarios of laminae preservation and proxy alignment (sea ice, open ocean, stratification as 2 paragraphs max each). 4.2.3. Physical parameter alignment (seasonal atmospheric and sea ice links to laminated records) 4.2.3 Adélie Land specific proxies (compared to other regions previous papers).

Or (2) Due to the extensive detail in the paper and the fact that the atmospheric case studies also sit awkwardly earlier on in the paper, I recommend you cut the paper in to two papers. This BG paper should concentrate uniquely on the diatom evaluation with the HBI, sea ice and elemental data and the PCA results, therefore focusing on

the important species that this evaluation highlights. A second paper should then be based on this analysis and its final findings, and thus provide the broader climatological interpretation in clear context to the case studies of atmospheric and physical studies. I think implementing this two paper approach will allow the true value of the comprehensive study to shine through what is currently a very heavy discussion with far too much detail and no clear synthesis by the end. I have provided an annotated copy to the authors with respect to grammatical changes and suggested text refinements.

5. Conclusion. The conclusion is far too broad and sweeping, and fails to clearly highlight the important findings of the two major themes of this study. In part, I think the lack of clarity here is symptomatic of the huge amount of information that has been covered in the discussion and, secondly, because the main aims are not re-addressed. I suggest a complete rewrite in context to how the authors wish to move forward in a future resubmission.

---

## Author Comment (AC1) · 3 Jun 2016

Dear Editor, Please find below the detailed response (marked by a bullet point) to the reviewer comments. We agree that referee's' suggestions allowed us to greatly improve the quality of the manuscript and the overall message of the paper. We have rethink the discussion part of the paper and notably clarified the discussion. We hope that the responses and improvements we will provide in this second revised version will convince you that our manuscript entitled "Sedimentary response to sea-ice and atmospheric variability over the instrumental period off Adélie Land, East Antarctica" by Campagne et al. should now be published in Biogeosciences. Yours sincerely, Philippine Campagne

  Anonymous Referee #1 General Comments Diatom has been used as a proxy of paleoenvironmental information based on large scale diatom ecological studies, whereas little is known about local-scale diatom ecology. The authors attempted to solve this issue namely understanding the relationship between diatom communities and local environmental conditions using the principal component analyses (PCA) and the Pearson correlation test.

Introduction Authors are needed to review previous studies on similar topics and discuss the problems remaining in the topic discussing in the paper. Some information is available in the text but in more comprehensive manner is requested.

• The Introduction clearly presents the problems and the frame of the manuscript to orientate the reader.

Environmental settings Authors should provide the detailed description of the quantitative criterion. Also rawdata of meteorological data need to be provided so that readers can evaluate them.

• This point is not very clear. Quantitative criteria about the meteorological data and sea-ice data can be found in section 3.3 and supplementary note S2. We believe that presenting raw meteorological data will uselessly make the manuscript heavier as they are not directly comparable to the sedimentary signal due to their high resolution. We believe that the transformed data presented in fig 3B are here enough.

Material and methods Information is lacking how the measurements of 210Pb chronology and biomarker analyses are conducted. Again raw data of 210Pb measurements and diatom assemblages are required either in the main text or in the Supplement. • The methodology of the 210Pb chronology is sufficiently described and referenced in section 3.1. Raw data are not necessary. Similarly, the methodology of diatom and biomarker analyses are sufficiently described and referenced in section 3.2. We believe that there is no need to repeat what has been lengthily described in other papers.

A question is also raised how to standardize different sort of data (diatom, proxies and meteorological parameters) in PCA?

• We first standardized our data by applying the basic formula, by substracting the mean of the distribution to the variable and dividing the result by the standard deviation. Once standardized, a normally distributed random variable has a mean of zero and a standard deviation of one, allowing different time series to be statistically confronted.

It seems that there is no need to do PCA using 67 diatom species data. Authors should conduct PCA using only abundant species.

• To be really exhaustive we worked in two steps. A first identification ofthe significant species/proxies was done through a complete dataset-based PCA. A second PCA was conducted between significant species/proxies, identified in the first PCA, and environmental parameters. This approach allowed us to surprisingly evidence that F. curta variations were not significant in our study (location, timescale). Following reviewers' advice, we decided to move the 'first PCA' in the supplementary material (Note S1) in order to simplify the scientific message.

Results and discussion Some re-organization of the section is required. The section 4.1 and 4.2 are too long, a part of these sections can be moved to the Supplement. It is appropriate to discuss the axis interpretation (subsection 4.2.1) first then discuss the relationship between sedimentary proxies (section 4.1). A part of the second paragraph of the section 4.1.1 should be moved to the Methods section.

• We here fully agree. The discussion part has been rewritten accordingly to both reviewers' comments. Most of old section 4 has been moved to the Method or Supplementary Material as requested. The discussion (section 5) is now separated in two sections. Section 5.1 presents the significance, behavior and relationships between the relevant proxies over the last 40 years, along with the differences between our new findings and their use in literature . More exhaustive information on principal and secondary proxies are now detailed in Note S1. Section 5.2 confronts instrumental

records with our sedimentary signals through statistical analyses in order to refine the ecological preferences of our relevant proxies at the regional scale. Section 5 has been shortened.

Figure The figures would be required for some editing. Sites discussed in the text should be added in the figure 1.

• We believe that presenting the different sites in the figure, that also mentioned wind and topographic features..., will uselessly make the figure heavier.

Also add the dominant direction of winds.

• We agree and we have changed Figure 1 according referee comments.

Photographs and stratigraphic columnar sections of core should be presented.

• We think this should be not necessary as the core is composed of diatom oozes through the all section. However, we could present in the Supplementary SCOPIX images of the core showing the sediment structure if the referee think this is necessary.

Figure 5 is too small to decipher and bigger figure is needed.

• We agree. This figure has been enlarged and improved.

Authors should describe details of figures such as the meaning of black arrows in the Figure 7?

• Black arrows from this figure have been removed. In other figures, black arrows have been described.

What does dashed line in the Figure 8 mean?

• Dashed lines have been described in this figure as length of the sea ice free season.

Moreover, authors should write environmental information (e.g. east winds, south winds) outside the graph to make the figure clear and concise.

• Environmental information quoted into the graph represent a secondary product of the data that are all described along their Y axes. As such, We kept the environmental information inside the graph as further visual support to the reader.

Supplement Some re-organization of the supplement text is required with "Results and Discussion" section. Authors should adjust the table size appropriate for publication. Table S2 is too large (continued 7 pages!!), and Table S3 is illegible for the small size.

• The Supplement Material has been re-organized as requested (see above). Tables have been improved.

Page 13, Line26: tri unsaturated -> tri-unsaturated

• Change has been done.

Page 14, Line 3: co occurrence -> co-occurrence

• Change has been done.

Figure 7: Add the unit of y-axis.

• Change has been done.

Figure S2: Add the unit of x-axis and y-axis.

• Change has been done.

Table S1: Add the description of each meteorological parameters to the table caption.

• We now describe meteorological parameters in the legend of the table.

  L. Armand (Referee 2) General Comments. This paper focuses on the understanding the diatom, elementaland isoprenoid records preserved in an _40 year old, annually laminated, interfacecore from the DurmontD'Urville Trough, Adelie Land, East Antarctica. The study attemptsto relate the changes in down core records of the preserved biological andchemical records to both prevailing regional atmospheric and sea-ice conditions for thesame time period assuming the laminations are linked.Although

the paper makes a significant advance to our understanding of the environmentalconditions of the Adélie –George V Land coastal region as is recorded in a 40year time series, the paper requires a major refocusing or dividing in to two papers sothat story is more succinctly and clearly delivered. At present it is very hard to readthrough and follow how the main aims are delivered on. It needs to be significantlyreorganised and cut back, which I strongly encourage all the co-authors to assist thefirst author in undertaking.

The results need to be extracted into their own section and subsections and thereforeseparated from the heavy discussion sections in which they are currently hidden.Much of the ecological discussion should be cut back and either placed in if relevantto the current understanding of climate interpretations in the Introduction or used tosupport/provide contrast to the results in a discussion section. For example: qualificationsof species groupings and/or resting spores/phases that have been applied priorto analysis should be found in the methods section (3.2) and not in the discussion.

Terminology needs careful attention (for example seasonal sea-ice zone = the areabetween the winter and summer sea-ice extremes; whilst marginal ice zone = the lowsea ice concentration typically found at/near the ice edge and can be either quite diffuseand expansive (and hundreds of kilometres wide) or quite narrow and less than a fewkilometres, usually depending on the winds). The use of a hyphen is required for seaice when used as an adjective (e.g. sea-ice edge, sea-ice concentrations, sea-iceextent) but is not used for the noun (e.g. Antarctic sea ice, polar sea ice). Hyphensshould also be used for sea-surface temperature (and salinity).

• We structured the revised version of the manuscript according to the reviewers' suggestions. More precisely, the Results are presented in section 4 and the Discussion in section 5. The manuscript has been drastically shortened and focused on the main scientific issue. Supporting information have been moved to the Supplementary Material.
Adherence to basic protocols in the identification of diatom species in the text shouldbe followed, where the first time that the species is mentioned in the text it should bewritten in full (Fragilariopsiscylindrus) and then the genus name can be abbreviatedthereafter (F. cylindrus), with exception to the start of a sentence.

• Although we tried to follow basic protocols in the identification of diatom species some errors seemingly persisted in the submitted manuscript. This has been checked thoroughly in the revised version.

Someone will need to check the English to capture minor errors that appear through-outthe paper and to add commas where required. I have sent an annotated copy of thepaper to the author for this purpose.

There are a lot of acronyms – this might want to be reconsidered, as to how necessary some are.

• We reduced the number of acronyms by discarding non important ones.

Specific Comments Abstract: Needs a rewrite to be more specific on the results iden-tifiedand more direct about what the climatic relationship may be (i.e does a species orlaminae relate to a oceanic, sea ice or atmospheric condition?)

1970-2010 = _40 years = 72.5cm long

• This point is not clear. It is obvious from the abstract, manuscript and fig 2 what is the time period covered by the core.

CE or AD ?? Editor might need to decide onwhat they prefer in this journal.

• We will change this point in the text following the editor decision.

Use of contrasting sea ice zone (SIZ vs Marginal IZ)are these zone what you really mean?

• We now use the right term in the right place.

[Figure]

What are the major elemental "abundances"(concentrations?) that are significant to this study? Can you be more specific?

• The important point here is not the elemental concentrations by themselves but how their variations are in line, or not, with climate/environmental changes as recorded by the instrumental data. Other major, minor and trace elements are positively or negatively related to environmental changes. We however presented here the most commonly used in the literature.

Resultsare not clearly identified or conveyed- what about the HBI signal? Last sentence shouldbe removed and replaced with a more specific statement on interpretation and impactresolved.

• The Results section has been improved.

1. Introduction I suggest adding two more short paragraphs (1) on relevant diatom-coastal interpretations taken from discussion 4.1 sections and (2) on current HBI inter-pretationsalso found in discussion 4.1 sections

• We believe that adding new paragraphs in the Introduction will uselessly make it heavier.

2. Environmental Sections 2.1 minor grammatical error (limit! limits)

• Corrected in the new version.

2.2. needs tighteningup on the English Sentence change suggestions. '. . . originates from brine rejection,during winter sea-ice formation. . .'

• Changed in the new version.

sea bottom ! sea floor?

• Changed in the new version.

'Adelie Land BottomWater is a major contributor to Antarctic Bottom Water'.

• Changed in the new version.

2.3 Sentence change suggestions'. . . coinciding with maximum wind speeds...'

• Changed in the new version.

2.4 'Sea-ice' conditions This sectionwould benefit significantly by having a figure/images illustrating the core's location incontext to the major sea-ice variations described.

• We agreed and integrated satellite images of sea ice conditions in Adélie Land in the Figure 1 in the new version.

2.5 Please describe more specifically inwhat way these studies are limited in time and low resolution (in reference to the state-ment: Most of these studies are either limited in time (refs) or are too low in resolution. . .) 2.5.1 to 2.5.3. I am not sure all the detail presented in these three subsections is necessary within the paper. You need to decide if these subsections are another study that needs to go in a separate paper altogether that you can refer to, or that you have completed a summary analysis (detailed in Supplementary section 1), which is then categorised by the three condition types that you apply as standard atmospheric conditions in this paper's analysis (potentially this could be summarised within one paragraph), or if the analysis itself is a major part of the study and the Case studies should be in the results section instead. At present it is too much information, and looks like results of a separate study and thus ultimately detracts from the paper's aim. The section needs a rethink in terms of what purpose it is serving in this paper.

• We agreed. In the new version, the 2.5 section has been moved to the Supplementary information, as we think this work helps, along with previous studies, to draw different scenarios of atmospheric pattern.

Some final comments on the case studies : it isn't clear on what time scales the associations have been based 1 year, 5 years, 20 years etc.

⇢ PCA between sedimentary and meteorological signals are based at 1 year tim-scale. We have add informations on it in the new version.

There are too many acronyms.What is AWS?

⇢ AWS=Automatic Weather Station, this accronym has been removed from the new version.

3 Materials and Methods 3.1 Should the word (CANBERRA) bein the text? Did you mean the analyses were undertaken at the Australian NationalUniversity in Canberra, Australia? I suggest writing the text in the past tense.

⇢ Canberra is the equipment. Analyses were performed at EPOC, University of Bordeaux.

3.2 There are two different references outlined in two sentences n this section as hav-ingbeen followed (Crosta and Koç 2007 and Crosta et al. 2004), this should be clari-fiedbetter. The references used in taxonomic identification should have been detailed here(even if in an appendix or supp. material) particularly given that some of the sea-icetaxa are not necessarily found in the Crosta works listed. Details defining the cate-gories,groupings of taxa or use of resting spores (RS) should have been documented-here rather than in the discussion sections. It may be worthwhile placing reference to Table 1 in this section given it is a summary of the published diatom ecological data.

⇢ This part has been clarified. We decided not to detail the groupings of taxa here as they result from the first PCA and are, therefore, already some results. We think that placing reference to Table 1, which is a summary, would make the table heavier.

Remove the third sentence, but retain the NOA61 usage, detail the type of microscope.

⇢ Change has been done.

Indicate where the data is can be accessed (in a proper public data repository for futureuse) particularly if you do not provide a sup. table with this information with

thepaper. How can anything be verified from this work?

• We agree and we will provide the data online.

Note here you indicate 70 speciesidentified and only 25 species are greater than 2% of total relative abundance, later youindicate under PCA there are 65 species that are used many with abundances less than1%. I would like to see a table summarising the diatom abundance data in the paperat the very least and related to full public data.

• We agree and added a table with diatom relative abundances.

3.4 Can youplease check all the dates (1978-2012; 1956-2011; 1979-2009) and make it clear whatseries you are using, or confined to, for the various instrumental data series given youhave from Pb dates indicating the core is 40 years old (1970-2010). I found this sectionvery difficult to follow any reasoning, when some years went beyond or started prior tothe core's actual record of preserved conditions.

• All dates are exact. We here provide the full range of extracted instrumental data for sea ice (1978-2012) and winds and air temperature (1956-2011). Instrumental and sedimentary data were standardized based on the mean value calculated on the 1979-2009 period, a period that covers instrumental data (AWS and satellite) along with sedimentary data.

• The separation of this chapter in two sub-chapters easies the reading.

Remove the reference to PCA here as you are only describing the data used not the analyses undertaken upon it later.

• Change has been done.

Why was the 40% sea-ice concentration threshold chosen as the difference be-tweensea ice covered and non-covered periods? Is this threshhold one that has been identifiedby the sea-ice observation community as a valid concentration along the coast ofAdélie Land for this sort of open or closed cover? I know that it makes sense out

in theoffshore SIZ, but wonder if that holds true close to the Adélie Coast. Please providea reference and/or images in Section 2.4 that help to support your choice. If it wasarbitrarily chosen to represent these conditions, as they occur in the offshore SIZ, thenjust say so.

• In the sea ice community, 15% represents the threshold between open waters and sea ice while 40% represents the threshold between unconsolidated and consolidated sea ice. Many studies have shown that diatoms, along with other organisms, develop early in the season when light returns and sea ice is still consolidated. We think that using 15% as a threshold, hence shortening the growing season to the open period, would strongly reduce (1) the representativity of the instrumental records to only few weeks and (2) the relationship between environmental conditions and the phytoplankton response. To note also that tested different thresholds (from 20 to 60%) and very similar results. We therefore think that the 40% value is a rather robust, biologically representative threshold.

3.5 Please make sure that you add the two identified points provided asreasons behind the PCA analyses to your introduction aims undertaken to answer yourhypothesis. Start a new paragraph with the text beginning with 'In a second PCA. . .'and then rewrite this being clearer about what you mean by 'significant sedimentarydata, in terms of population and ecological preferences' and 'interpolated at one year'(did you mean averaged over one year?). Delete the last half of the last sentence orweave that into the Introduction (, to support. . ..).Should there have been a material and methods section also focused on the elementalanalyses undertaken?? Where is this data located publically?

• This part has been corrected.

4. Results and discussion. The majority of text in the first part of Section 4 could beused in the Introduction; much is neither results nor discussion. Also 'George V Land'typo fix.4.1 Should be re-labelled as the RESULTS section. In here you really

should place theresults of each analysis that has been covered in the order previously delivered in thepaper under methods and materials. Tables, Figures and reference to Supplementarymetadata should be identified in the relevant results subsections. Having read throughthe rather hefty sub-sections of section 4. I have two suggestions on how you maywant to reconfigure the paper to provide a clearer message.

• We agree with the referee 2, and the result and discussion parts have been rewritten according to the reviewers' suggestions.

• In the new version we chose to focus only on the relationships between significant sedimentary proxies and environmental parameters. Significant means that their abundances and variations are large enough to be statistically recorded by the PCA. Fragilariopsis curta conversely present low amplitude changes, unrelated to any other diatom species in core DTCI2011, and is therefore not statiscally significant despite a mean relative abundance of 20%. In the new version, results are described in section 4. We moved the first PCA into the Supplementary Material instead of merely abandoning it as we think it is an important statistical element to define which proxies should be here selected. The discussion (section 5)is now separated in two sections. Section 5.1 presents the significance, behavior and relationships between the relevant proxies over the last 40 years, along with the differences between our new findings and their use in literature . More exhaustive information on principal and secondary proxies are now detailed in Note S1.Section 5.2 confronts instrumental records with our sedimentary signals through statistical analyses in order to refine the ecological preferences of our relevant proxies at the regional scale. Section 5 has been shortened.

5. Conclusion. The conclusion is far too broad and sweeping, and fails to clearlyhighlight the important findings of the two major themes of this study. In part, I thinkthe lack of clarity here is symptomatic of the huge amount of information that has beencovered in the discussion and, secondly, because the main aims are not re-addressed. I suggest a complete rewrite in context to how the authors wish to move forward in a future resubmission.